# Evaluating the dependence structure of compound precipitation and wind speed extremes

Jakob Zscheischler[1,2,3], Philippe Naveau[4], Olivia Martius[1,5,6], Sebastian Engelke[7], and Christoph C. Raible[1,2]

[1]Oeschger Centre for Climate Change Research, University of Bern, Bern, Switzerland
[2]Climate and Environmental Physics, University of Bern, Sidlerstrasse 5, 3012 Bern, Switzerland
[3]Department of Computational Hydrosystems, Helmholtz Centre for Environmental Research – UFZ, Leipzig, Germany
[4]Laboratoire des Sciences du Climat et de l'Environnement, Gif-sur-Yvette, France
[5]Institute of Geography, University of Bern, Bern, Switzerland
[6]Mobiliar Lab for Natural Risks, University of Bern, Bern, Switzerland
[7]Research Center for Statistics, University of Geneva, Geneva, Switzerland

**Correspondence:** Jakob Zscheischler (jakob.zscheischler@climate.unibe.ch)

**Abstract.** Estimating the likelihood of compound climate extremes such as concurrent drought and heatwaves or compound precipitation and wind speed extremes is important for assessing climate risks. Typically, simulations from climate models are used to assess future risks, but it is largely unknown how well the current generation of models represents compound extremes. Here, we introduce a new metric that measures whether the tails of bivariate distributions show a similar dependence structure across different datasets. We analyse compound precipitation and wind extremes in reanalysis data and different high-resolution simulations for central Europe. A state-of-the-art reanalysis dataset (ERA5) is compared to simulations with a weather model (WRF) either driven by observation-based boundary conditions or a global circulation model (CESM) under present-day and future conditions with strong greenhouse gas forcing (RCP8.5). Over the historical period, the high-resolution WRF simulations capture precipitation and wind extremes and their response to orographic effects more realistically than ERA5. Thus, WRF simulations driven by observation-based boundary conditions are used as a benchmark for evaluating the dependence structure of wind and precipitation extremes. Overall, boundary conditions in WRF appear to be the key factor in explaining differences in the dependence behaviour between strong wind and heavy precipitation between simulations. In comparison, external forcings (RCP8.5) are of second order. Our approach offers new methodological tools to evaluate climate model simulations with respect to compound extremes.

*Copyright statement.* TEXT

## 1  Introduction

Compound extremes such as co-occurring drought and heat or compound precipitation and wind extremes can have substantial impact on the natural environment and human systems that often exceeds impact caused by a single extreme (Zscheischler et al.,

2014; Raveh-Rubin and Wernli, 2015; Martius et al., 2016; Sippel et al., 2018). Over the recent years a number of compound extremes have been investigated. For instance, several studies have analysed the dependence between storm surge and heavy precipitation (Wahl et al., 2015; Zheng et al., 2013; Bevacqua et al., 2019) or extreme runoff (Ward et al., 2018; Hendry et al., 2019) to estimate the risk of compound flooding in coastal areas. Compound droughts and heatwaves have been studied for different regions and varying temporal scales (Mazdiyasni and AghaKouchak, 2015; Zscheischler and Seneviratne, 2017; Manning et al., 2019; Sutanto et al., 2020; Zscheischler and Fischer, 2020). The occurrence rate of compound precipitation and wind extremes has been estimated for the Mediterranean region (Raveh-Rubin and Wernli, 2015), Europe (De Luca et al., 2020) and at the global scale (Martius et al., 2016). Other studies have investigated the co-occurrence of hot days and hot nights (Wang et al., 2020) or the co-occurrence rate of heavy precipitation and snow melt to estimate the risk of rain-on-snow events (Musselman et al., 2018; Poschlod et al., 2020). Such a quantification of the occurrence rate of compound extremes is important for assessing the risk of associated impacts today and in the future. Most of the above studies identify compound extremes by thresholding the contributing variables to quantify the occurrence of compound extremes and changes associated with climate change. However, the dependence structure in the tails only is rarely investigated. Due to the rarity of compound extremes, a large number of samples is required to obtain robust estimates, making it difficult to rely solely on observational data (Ridder et al., in press).

Large ensemble simulations (Deser et al., 2020) offer an opportunity to estimate future changes in the occurrence of compound events without running into data limitations (Poschlod et al., 2020; Champagne et al., 2020). However, such projections need to be interpreted with care as it is often largely unknown how well the employed models represent observed compound events (Musselman et al., 2018), and differences might be large between models. Climate models are regularly evaluated based on their ability to represent well-known processes in the climate system as well as predominantly univariate comparisons with key climate variables (Flato et al., 2013) though some multivariate metrics have been explored (Sippel et al., 2017). Yet little is known about the ability of climate models to capture observed occurrence rates of compound extremes (Zscheischler et al., 2018), a challenging task primarily for two reasons. First, due to their rarity, a robust quantification of the likelihood of compound extremes requires large amounts of data, thus making it difficult to establish "ground truth" for many applications. Second, suitable metrics for evaluating multivariate extremes have not been widely tested and applied in a climate context. Such metrics, however, are essential to assess how well models represent compound events, in particular to assess future risks (Zscheischler et al., 2020). When observational data are scarce, process-based model simulations (Couasnon et al., 2020) and reanalysis data (Martius et al., 2016) can be employed to extend or replace purely observational datasets.

To date, model-data comparisons related to compound extremes have been conducted to a very limited extent, often relying on simplifying assumptions and typically confined to precipitation and temperature. For instance, a high likelihood of compound hot and dry summers has been linked to a strongly negative correlation between summer temperature and precipitation (Zscheischler and Seneviratne, 2017). While there is generally a good agreement with respect to this correlation between climate models and observation-based datasets in the northern hemisphere, there is strong disagreement in the southern hemisphere, for which the models show a much stronger dependence. This finding may suggest that climate models overestimate dependence between summer temperature and precipitation. However, this discrepancy may also be related to the way gridded

observation-based datasets are assembled. In particular, for locations without an active measurement station nearby, the mean

seasonal cycle is often used to fill gaps in the observational networks (e.g. Mitchell and Jones, 2005). This approach reduces the strength of co-variability between temperature and precipitation in poorly sampled regions, which are mostly in the southern hemisphere. Thus, assessing the ability of climate models to represent compound events may reveal underappreciated limitations in gridded observation-based datasets. We are not aware of studies so far that have evaluated the dependence between precipitation and wind speed.

In this study we focus on compound precipitation and wind extremes, which can have severe socio-economic impacts including human fatalities, impaired critical infrastructure and economical damage (Fink et al., 2009; Lin et al., 2010; Liberato, 2014; Raveh-Rubin and Wernli, 2015; Martius et al., 2016). We investigate differences in the occurrence of compound precipitation and wind extremes for different datasets over a region in central Europe around the Alps. To this end, we introduce a new measure that assesses dissimilarity between the tails of bivariate distributions. We study an experimental design with two

factors. The first factor is the type of boundary conditions in a high-resolution regional weather model, either from reanalysis or a global circulation model. The second one corresponds to the effect of different climate forcing, between today and the future under a high-emission scenario. Our object of study under this design is the dependence between heavy precipitation and strong wind in winter over central Europe. In addition, comparisons with a state-of-the-art reanalysis product are implemented.

## 2  Data

We use daily precipitation sums and daily maximum wind speed in the extended winter (November-March) from one reanalysis product and three model simulations over a period of 20 years. The employed reanalysis product is the ERA5 data (Copernicus Climate Change Service (C3S), 2017) where we use the period 1980 to 1999 CE. This reanalysis is generated with an updated numerical weather prediction model and data assimilation system compared to the prior product ERA Interim (Dee et al., 2011) and integrates additional data sources. The data is available at resolution of roughly 30 km (spectral resolution of T639), 137

vertical levels and hourly output.

The three simulations are performed with the Weather Research and Forecasting (WRF) model (Skamarock et al., 2008) which is forced with boundary conditions from (i) ERA Interim (Dee et al., 2011) (ERAI-WRF), (ii) a period of free-running global climate simulation for present day (CESM-WRF) and (iii) a period covering the end of the 21st century under the Representative Concentration Pathway 8.5 (CESM-WRF-fut, a high-emission scenario). The global climate simulation is performed

with the Community Earth System Model CESM (Hurrell et al., 2013) for the period 850 to 2100 CE. Details on the setting are described in Lehner et al. (2015) and Raible et al. (2018). In this study we use the periods 1980 to 1999 CE as present day and 2080 to 2099 CE as future.

The periods of the global simulations and the ERA Interim period (1980 to 1999 CE) are dynamically downscaled with WRF in version 3.5. WRF is vertically discretised in 40 terrain-following eta-coordinate levels. The horizontal resolution

of the four two-way nested domains (Fig. 1) are 54, 18, 6 and 2 km, respectively. The innermost domain covers the box $[4.75°E, 15.25°E] \times [43.25°N, 48.75°N]$ and is used in this study exclusively. The setup is described in more detail in Gómez-

Navarro et al. (2015, 2018) and Messmer et al. (2017, 2020). Important for this a study is that the convection parameterisation is disabled for the simulations at 6 km and 2 km resolution; at these scales the model is convection-permitting. This is an important step in improving the simulation of precipitation, though still some problems remain (Ban et al., 2014). For simulating wind adequately, the setting of the planetary boundary layer parameterisation is key. We use a modified version of the fully non-local scheme developed by Hong and Lim (2020), which specifically treats effects of the unresolved orography (Jimenez and Dudhia, 2012). For the ERAI-WRF simulation we allow analysis nudging of wind, temperature and humidity above the planetary boundary layer, in order to stay close to large-scale behavior of the reanalysis data (Gómez-Navarro et al., 2015). For the two simulations driven by CESM, nudging is omitted to allow the regional model to correct potential systematic biases of the CESM (e.g., a too strong zonal atmospheric circulation in the mid latitudes (Bracegirdle et al., 2013)). The WRF output is provided in hourly resolution.

We remap the original hourly data to a common regular spaced grid with 0.25° spatial resolution using conservative remapping and subsequently compute daily precipitation sums and daily wind speed maxima. The 0.25° spatial resolution was chosen as it is closest to the original resolution of the ERA5 reanalysis data. Note however, that all WRF simulations are run on a much higher convection-resolving resolution. The explicit resolution of convection and a much higher resolution of the topography may result in a more accurate representation of the dependence between precipitation and wind extremes in the simulations than in ERA5. We further note that mean wind speed in ERA5 generally decreases with elevation (Fig. 2a), which is the opposite behaviour of what is the expected behaviour of the response of wind speed to elevation from observations (Graf et al., 2019; Telesca et al., 2020) and what is modelled by WRF (Fig. 2b). The discrepancy in mountainous regions between reanalysis data and observations with respect to wind speed is also evident in other reanalysis datasets such as ERA Interim (Jones et al., 2017), which is the predecessor of ERA5. In contrast, WRF has been shown to simulate wind speed reasonable well also in mountainous terrain (Stucki et al., 2016). For these reasons — WRF better resolves cloud processes, the topography and wind speed, ERA5 misrepresents wind speed gradient with elevation — we use ERAI-WRF as the reference for all analyses.

## 3 A measure for evaluating compound extremes

### 3.1 Measuring tail dependence

The extreme values of a univariate random variable can be analyzed with tools from extreme value theory (Embrechts et al., 1997; Coles, 2001; Katz et al., 2002; Naveau et al., 2020). For multivariate random vectors, the dependence between the largest values in the components becomes important (Davison and Huser, 2015; Huser and Wadsworth, in press; Engelke and Ivanovs, in press).

We quickly review the concept of bivariate asymptotic tail dependence and independence (Ledford and Tawn, 1997; Poon et al., 2003). Two variables $X_1$ and $X_2$ with cumulative distribution functions $F_1$ and $F_2$, respectively, are asymptotically dependent if

$$\chi = \lim_{q \to 1} \mathbb{P}(F_1(X_1) > q \mid F_2(X_2) > q) \in (0,1],$$

and asymptotically independent otherwise (i.e., if $\chi = 0$). The coefficient $\chi$ is called extremal correlation and represents, after transforming $X_1$ and $X_2$ to the uniform scale, the probability of one variable being extreme given that the other one is extreme. Note that two variables can be dependent at normal levels but asymptotically independent in the extremes, as in the case for a bivariate Gaussian distribution (Sibuya, 1960). To fine tune the rate of decay towards the asymptotically independent case ($\chi = 0$), the residual tail dependence coefficient $\bar{\chi}$ contains additional information (Coles et al., 1999):

$$\bar{\chi} = \lim_{q \to 1} \frac{\log(\mathbb{P}(F_1(X_1) > q)\mathbb{P}(F_2(X_2) > q))}{\log \mathbb{P}(F_1(X_1) > q, F_2(X_2) > q)} - 1 \ \in [-1, 1].$$

$\bar{\chi}$ is equal to 1 for asymptotically dependent variables, while for asymptotically independent variables $\bar{\chi}$ indicates if $X_1$ and $X_2$ are positively ($\bar{\chi} > 0$) or negatively ($\bar{\chi} < 0$) associated in their extremes. Thus, the pair of coefficients ($\chi, \bar{\chi}$) summarizes the tail dependence structure of $X_1$ and $X_2$.

Because both coefficients $\chi$ and $\bar{\chi}$ are defined as a limit value, a usual way to analyze the behaviour of a bivariate tail dependence structure between two variables is to compute empirical estimates for varying threshold levels $q$ and then visually inspect their behaviour as $q \to 1$. We estimate $\chi$ and $\bar{\chi}$ with the function taildep from the R package extRemes (Gilleland and Katz, 2016).

To estimate $\chi$ empirically we use a high quantile for which a reasonable large number of data are available. For these reasons we generally estimate $\chi$ at $q = 0.95$. Heavy precipitation events and extreme winds that lead to large damages can be linked through storms or foehn events across neighboring locations and with a lag of several days. To take this aspect into account, we estimate $\chi$ using a local block maxima approach, which is motivated by (Ferreira and de Haan, 2015). We thus first compute the daily precipitation and wind speed maxima for varying block sizes ranging from $0.25°$ (approximately 20-30 km) to $1.75°$ (that is, maximum 3 grid points in any direction, or 100-200 km) and up to 5 days (i.e., maximum 2 days before and after the day of interest).

We further assess whether estimates of $\chi$ are significantly different from 0. To this end, we bootstrap the data by randomly shuffling the temporal order of one variable to break the dependence structure. The coefficient $\chi$ is then estimated as above. Estimates of $\chi$ are considered significantly different from 0 if they are larger than 95% of the bootstrapped estimates.

## 3.2 Measuring differences in bivariate extremal dependence structures

Classical tail coefficients like $\chi$ are informative summaries to assess the extremal dependence between two univariate random variables, say $X_1$ and $X_2$, but they cannot quantify the difference between extremal dependence between two **bivariate** random vectors, say $\boldsymbol{X}^{(1)} = (X_1^{(1)}, X_2^{(1)})$ and $\boldsymbol{X}^{(2)} = (X_1^{(2)}, X_2^{(2)})$. For example, a $\chi^{(1)}$ can be computed between heavy precipitation and strong winds computed from one dataset, e.g. ERA5, and compared to a $\chi^{(2)}$ for a second dataset, e.g. ERAI-WRF. But it would also be very convenient to have a single number to tell us if the extremal dependence between these two bivariate random vectors are different, and if so, by how much. Recent work by Naveau et al. (2014) showed the well-known Kullback–Leibler (KL) divergence used in signal processing can be tailored to the framework of extreme value theory. The approach has been applied to cluster climate data according to their bivariate extremal behaviour (Vignotto et al., submitted). However, to our knowledge, multivariate extremal divergence measures have never been applied to the analysis of compound weather

and climate events. By complementing tail coefficients, this new tool could shed new lights on the joint behavior of heavy precipitation and strong winds across our different datasets.

The KL divergence is defined on marginals which are normalized to standard Pareto distributions. A risk function $r : \mathbb{R}^2 \to \mathbb{R}$ is used to describe the extreme region in each one of the bivariate distributions. There are different choices for the risk function. Taking the sum or the maximum give similar results for asymptotically dependent data. In addition, the minimum covers also asymptotic independence. The sum is defined as $r(\boldsymbol{x}) = x_1 + x_2$, the minimum as $r(\boldsymbol{x}) = \min(x_1, x_2)$, $\boldsymbol{x} = (x_1, x_2)$. Hence, we consider as extreme points those for which the sum (or minimum) of the components exceeds a given high quantile $q_u^{(j)}$ of $r(\boldsymbol{X}^{(j)})$ corresponding to an exceedance probability $u \in (0,1)$, $j = 1, 2$. Varying the threshold $q_u^{(j)}$ alters the extremal region of interest. For each of the two bivariate distributions, the set $A^{(j)} = \{r(\boldsymbol{x}) > q_u^{(j)}\}$, $j = 1, 2$, is partitioned into a fixed number $W$ of disjoint sets $A_1^{(j)}, \dots, A_W^{(j)}$.

For two random samples $\boldsymbol{X}_1^{(1)}, \dots, \boldsymbol{X}_n^{(1)}$, and $\boldsymbol{X}_1^{(2)}, \dots, \boldsymbol{X}_n^{(2)}$, from the distributions $\boldsymbol{X}^{(1)}$ and $\boldsymbol{X}^{(2)}$, the empirical proportions of data points belonging to set $A_W^{(j)}$ is computed as

$$\widehat{p}_w^{(j)} = \frac{\#\left\{i : \boldsymbol{X}_i^{(j)} \in A_w^{(j)}\right\}}{\#\left\{i : r\left(\boldsymbol{X}_i^{(j)}\right) > q_u^{(j)}\right\}}, \quad w = 1, \dots, W.$$

The difference between the extremal behaviours of the two distributions can then be measured as the KL divergence between the two multinomial distributions defined through these proportions, i.e.,

$$d_{12} = D(\boldsymbol{X}^{(1)}, \boldsymbol{X}^{(2)}) = \frac{1}{2} \sum_{w=1}^{W} \left( (\widehat{p}_w^{(1)} - \widehat{p}_w^{(2)}) \log(\widehat{p}_w^{(1)}/\widehat{p}_w^{(2)}) \right). \tag{1}$$

Note that this divergence is symmetric and since it is a non-parametric statistic it does not require additional model assumptions. Equation (1) contrasts differences among extremal dependence structures, both for asymptotically dependent and asymptotically independent data. The number of partitioning sets $W$ is a free parameter. If it is chosen too high, many sets will be empty, resulting in an undefined KL divergence. If it is too small, only a rough summary is computed but not really an estimate of tail dependence. We chose $W = 5$ in this study based on the simulation study shown in Appendix A. Under suitable assumptions the statistic $d_{12}$ follows a $\chi^2(W-1)$ distribution in the limit as the sample size goes to $\infty$, which allows us to estimate whether distances are significantly different from 0.

The approach is illustrated in Figs. 3 and 4. Figure 3 shows daily precipitation sums and maximum wind speed at grid point $9°E$, $46.75°N$ on the original scale (a, d), and with margins normalized to exponential scale (b, e) and to standard Pareto distributions (c, f) for ERAI-WRF (a-c) and CESM-WRF (d-f). The shown grid point reaches the highest $\chi$ at $q = 0.95$ in the ERAI-WRF simulation. The colors in all subpanels and the dashed lines in Fig. 3c and f highlight the three disjoint sets $A_1^{(j)}, A_2^{(j)}$ and $A_3^{(j)}$, respectively (see above). At the exponential scale moderate and large extremes can be seen well whereas at the Pareto scale only very extreme values can be identified easily visually. Figure 4 illustrates $\chi$ (a) and $\bar{\chi}$ (c) for the distributions of the two simulations and the divergence based on Eq. (1) with "sum" (b) and "min" (d) as the risk function, including 95% confidence intervals of the empirical estimates. The estimates of $\chi$ and $\bar{\chi}$ start to diverge somewhat for $q > 0.8$, suggesting different tail behavior (uncertainty ranges are estimated based on the R function chiplot from the package evd (Stephenson,

2002)). This impression is confirmed by the estimates of the KL divergence: For most thresholds $u > 0.5$ and both choices of the risk function the KL divergence is outside of the $95\%$-quantile of the limiting $\chi^2(W-1)$ distribution of the statistic $d_{12}$ under the null hypothesis of equal tail dependence structures. This means that we can conclude that the two distributions have significantly different tail behaviour.

Note that in the bivariate case, a simple approach to quantify the difference in tail dependence would be the difference between $\chi^{(1)}$ and $\chi^{(2)}$. However, for two distributions with the same $\chi$ coefficient but different dependence structure, it is impossible to distinguish the two cases. In a way, $\chi$ only focuses on the "diagonal". Furthermore, while in this work we focus on the bivariate case, the KL divergence defined by Eq. (1) could be easily implemented with higher dimensions $d = 3, 4, \ldots$, because it is only based on counting points in different subsets. In contrast, using $\chi$, the number of pairs will increase rapidly with the dimension $d$. In addition, $\chi$ coefficients will only capture pairwise dependencies.

We investigate how well different simulations represent the bivariate tail behaviour of daily precipitation sums and wind speed maxima in winter by comparing ERA5, CESM-WRF and CESM-WRF-fut against ERAI-WRF with the divergence as defined in Eq. (1) based the maxima over the spatio-temporal blocks that maximize tail dependence $\chi$ at $q = 0.95$. Using local block maxima ensures that $\chi$, $\bar{\chi}$ and the KL divergence measure very extremal upper tail behavior. Note however, that this approach leads to different block sizes depending on the location, which makes a direct comparison in space difficult. For the computation of the KL divergence (Eq. (1)) we use $u = 0.9$ and "sum" as the risk function. We further perform a sensitivity test using $u \in \{0.8, 0.85, 0.9, 0.95\}$. Furthermore, the marginals have been transformed into Pareto scale through ranking. The choice of marginal transformation only has a minor influence on the KL divergence (see Appendix A).

## 4    Results

We first present a simple correlation analysis based on Spearman's rank correlation coefficient. Daily precipitation sums and maximum wind speed are generally strongly correlated in winter in most areas of the study domain except in the northwest of Italy (Fig. 5). All model simulations show a relatively consistent pattern, whereas ERA5 shows negative correlations at the southern slopes of the Alps along the northwestern Italian borders (Fig. 5b). Most correlations are significant ($\alpha = 0.05$).

When considering only the dependence in the tails based on $\chi$ and including a spatial and temporal neighborhood, the spatial patterns look quite different (Fig. 6). The WRF simulations show a highly heterogeneous picture with strong local variations, with generally strong dependence over most parts of the Alps and close to the Adriatic coast and weak dependence otherwise (Fig. 6a, c, d). Overall, ERAI-WRF shows slightly higher tail dependence compared to the WRF simulations driven by CESM. In contrast to the WRF simulations, in ERA5 tail dependence varies rather smoothly in space, with higher values in northeast Italy and along the eastern border of France (Fig. 6b).

The block sizes that attain the maximum tail dependence $\chi$ for precipitation and wind extremes for each pixel are shown in Fig. 7. On average for $75\%$ of the pixels, the maximum is attained with no temporal lag. In contrast, there seems to be a shift in space, as maxima tend to co-occur in neighboring grid points: block sizes with larger than minimal ($0.25°$) spatial extent occur on average in $60\%$ of all locations (lighter colors in Fig. 7). This means that extremes in daily precipitation sums and

wind extremes tend to occur on the same day but potentially not exactly at the same location but with some distance apart. In particular in the south of the Alps but also in some regions north of the Alps, this distance is 1.75°, or about 100-200 km (very light colors in Fig. 7). The strongest agreement of the dependence patterns exist between CESM-WRF and CESM-WRF-fut, which agree for half of the locations in the maximizing block size. In contrast, the agreement is 29% between ERAI-WRF and ERA5, and 39% between ERAI-WRF and CESM-WRF. Note that grid points at the boundaries cannot attain maxima with block sizes larger than one grid point as no data values are available outside the study domain.

The tails between winter daily precipitation sums and wind speed maxima show a significantly different dependence structure between ERAI-WRF and CESM-WRF in 46% of all grid points, mostly in Switzerland and in the north of the study domain but also in many regions in northern Italy (Fig. 8a). The percentage of grid points with significantly different tail behaviour is slightly higher for the comparison of ERAI-WRF and ERA5 (49%) though in this case most of the differences occur in grid points located along a wide diagonal band from south west to north east through the entire study domain (Fig. 8b). Interestingly, the comparison of ERAI-WRF with CESM-WRF-fut results in only 36% pixels with significantly different tail behaviour (Fig. 8c). Thus, CESM-WRF-fut agrees better with ERAI-WRF with respect to the tail behaviour than CESM-WRF and ERAI-WRF. Finally, only 18% of pixels show significantly different tail behaviour when comparing CESM-WRF and CESM-WRF-fut (Fig. 8d), indicating the pair with the largest number of grid points where no significant difference in the tail behavior could be found. The numbers of grid points with significantly different tail behaviour depends somewhat on the threshold $u$ and generally decrease with increasing extremeness (that is, increasing $u$) but the differences between the different pairwise comparisons remains similar (Table 1). In particular, the differences between ERAI-WRF and CESM-WRF and between ERAI-WRF and CESM-WRF-fut are generally larger than the differences CESM-WRF and CESM-WRF-fut, indicating that the main finding, namely that boundary conditions in WRF appear to be the key factor in explaining differences in the dependence behaviour between wind and precipitation extremes, is robust for different parameter values of the difference measure.

## 5   Discussion

We have introduced a new metric for comparing tail dependence structures between wind and precipitation extremes in reanalysis data and weather model simulations. In our WRF simulations, the type of boundary conditions, either ERAI or CESM, appears to have a stronger effect on the coupling between high wind and heavy precipitation than the change of external forcing (present-day and future) in CESM (Fig. 8). This suggests that the studied dependence structures between the tails of precipitation sums and wind speed maxima in winter are a rather robust feature of the combination of models (boundary conditions plus high-resolution weather model) and thus also somewhat determined by the boundary conditions. In consequence this also means that here we are probably detecting rather stable dynamical features that are largely independent of strong external forcing such as (much) higher mean temperatures. Because the model setting determines the dependence structure, sampling uncertainties in this dependence, for instance to robustly assess risks under future climate conditions, would require a range of different climate and weather model combinations.

The employed block maxima approach (Fig. 6) has the effect that precipitation and wind extremes are considered together even if they might occur some distance apart in either time or space. This is to ensure that extremes in wind and precipitation are considered together if they emerge from the same atmospheric processes (e.g. Foehn). At the same time, the block maxima approach can help diagnose why datasets differ in their tail dependence structure of precipitation and wind extremes, for instance if the spatio-temporal blocks for which extremes are attained differ strongly.

Regarding the optimal spatial and temporal lags between wind and precipitation extremes there is generally a good agreement that along the southern slopes of the Alps the dependence is maximized for precipitation and wind extremes occurring on the same day and up to 1.75° apart (lightest blue in Fig. 7), which could be related to Foehn events that lead to heavy precipitation north of the mountain range and extreme winds on the southern slopes or vice versa. Indeed, heavy precipitation events on the Alpine southside are often related to high moisture transport ahead of cold fronts that is associated with moderate winds that

are not as strong as potential Foehn gusts on the Alpine north side (Panziera and Germann, 2010).

Most heavy precipitation events in the investigation domain in winter are associated with extratropical cyclones. Within extratropical cyclones, wind speed maxima and precipitation maxima are often linked to fronts and conveyor belts (Parton et al., 2010; Catto and Pfahl, 2013; Pfahl et al., 2014; Pantillon et al., 2020) and this may result in co-located extremes. However, important modulations of both extreme wind and precipitation patterns by the local complex orography are to be

expected (Whiteman, 2000; Barry, 2008) and such local Foehn effects, channelling effects, or flow blocking and many more might be captured by the high resolution WRF simulations but not in ERA5.

Overall, ERA5 shows quite a different behavior in Spearman's rank correlation (Fig. 5) and simple tail dependence $\chi$ (Fig. 6) compared to the high-resolution weather model simulations. Spatial patterns are much smoother, probably related to the much coarser spatial resolution (30 km compared to the original 2 km in the WRF simulations). Furthermore, wind speeds over high

mountains are unrealistic, as they decrease with height rather than increase (Fig. 2). These limitations render ERA5 unsuitable as a benchmark for the tail dependence between precipitation and wind extremes in the Alpine area with its complex orography. Presently, homogenized gridded wind observations of good quality are not available for this region. Therefore, driving a well-calibrated high-resolution weather model with observation-based boundary conditions is currently the best benchmark to study compound wind and precipitation extremes.

We would like to note that in our setup ERAI-WRF is nudged to the driving reanalysis ERA Interim. The reason for this is that the simulation should stay close to large-scale behavior of the reanalysis data. As mentioned in the methods section, we only use wind, temperature and humidity above the planetary boundary layer and the nudging is not strong. Nevertheless, the behavior of extremes might be changed due to the modification of the dynamical equations to some extent, but we think that this effect is minor. Furthermore, precipitation is not nudged.

Evaluating how well models represent tail dependencies may help selecting those models that are fit for purpose (Maraun et al., 2017) regarding the analysis of compound events (Zscheischler et al., 2020) for a range of different event types (Ridder et al., in press). In particular, when the interest lies in the simulation of impacts, the approach may help decide when multivariate bias adjustment approaches would need to be employed (François et al., 2020), as univariate bias adjustment might increase biases in impacts that depend on multiple correlated drivers (Zscheischler et al., 2019).

## 6 Conclusions

Evaluating the ability of climate models to represent the likelihood of compound climate extremes is important for well-informed climate risk assessments. In this study we investigated differences in the tail behaviour of precipitation and wind extremes in winter between different weather model simulations and a reanalysis dataset for a region in central Europe. Employing a new metric to measure differences in tail behaviour of bivariate distributions, we found that simulations of the same model pair with different external forcing conditions (climate change conditions) differ less than simulations for present-day conditions with different boundary data. Our results further suggest that reanalysis data are not suitable as a benchmark for the analysis of compound precipitation and wind extremes over complex terrain such as the Alps. Overall, differences between model simulations (different boundary conditions and weather/climate models) can be substantial. Our results suggest the climate impact modelling needs to take uncertainties related to the simulation of compound extremes into account to provide robust risk assessments for today and the future.

*Data availability.* ERA5 data are available from the ECMWF website: https://www.ecmwf.int/en/forecasts/datasets/reanalysis-datasets/era5. The output from the WRF simulations are very large data files and are available from Christoph Raible (christoph.raible@climate.unibe.ch).

## Appendix A: Determining $W$

We simulated $n = 2000$ samples of $\mathbf{X}^{(1)}$ and $\mathbf{X}^{(2)}$ of the outer power Clayton copula, which is in the domain of attraction of the logistic extreme value distribution. We chose the parameters such that the limiting $\chi$ coefficients are 0.4 and 0.55, that is, one model with weaker and one with stronger dependence, respectively. Using the KL divergence for a probability threshold of $u = 0.9$, we compare the samples of $\mathbf{X}^{(1)}$ and $\mathbf{X}^{(2)}$ for the dependence settings weak/weak, strong/strong and weak/strong and plot in each case the probability of rejecting the null hypothesis of equal tail dependence structures. Note that the former two cases are in line with the null hypothesis, whereas the latter case does not satisfy the null hypothesis. We do the experiment both for known margins and for empirically normalized margins, and for different numbers of sets $W$ in the KL divergence statistic.

Figure A1 and A2 show the Type I error of rejecting the null hypothesis in the case where we have the same tail dependence based on 500 repetitions of the simulation based on empirical ranking of the marginals and using the true marginals, respectively. For both normalizations the significance level of 5% is in general well attained throughout all numbers of sets. The figures also contain the power of the test when the tail dependence structures are different. After $W = 5$ the power stabilizes and it seems to decrease slightly when the number of sets is chosen to large. We therefore use $W = 5$ throughout the manuscript. Note that this is only one particular simulation setup and the results on the optimal number of sets can change depending on sample size and strength of tail dependence.

*Author contributions.* J.Z. and P.N. conceived the idea and study design. P.N. and S.E. developed the code for the new metric. C.C.R. provided the model simulations. O.M. helped with the interpretation of the results. J.Z. performed all analysis, created all figures and wrote the first draft. All authors contributed substantially to the writing and revising of the manuscript.

*Competing interests.* The authors declare that they have no competing interests.

*Acknowledgements.* This research was supported by a Short-Term Scientific Mission from the European COST Action DAMOCLES (CA17109). We thank Martina Messmer for creating Figure 1. J.Z. acknowledges financial support from the Swiss National Science Foundation (Ambizione grant 179876) and the Helmholtz Initiative and Networking Fund (Young Investigator Group COMPOUNDX, Grant Agreement VH-NG-1537). C.C.R is supported by the Swiss National Science foundation (grant: pleistoCEP – no. 200020_172745). The CESM and WRF simulations were performed on the supercomputing architecture of the Swiss National Supercomputing Centre (CSCS, Lugano, Switzerland. O.M. is supported by the Swiss National Science Foundation (grant 178751). Part of Philippe Naveau's research was supported by the FRAISE-LEFE-MANU grant and the french Agence National de la Recherche throughout the ANR-Melody and ANR-TRex.

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

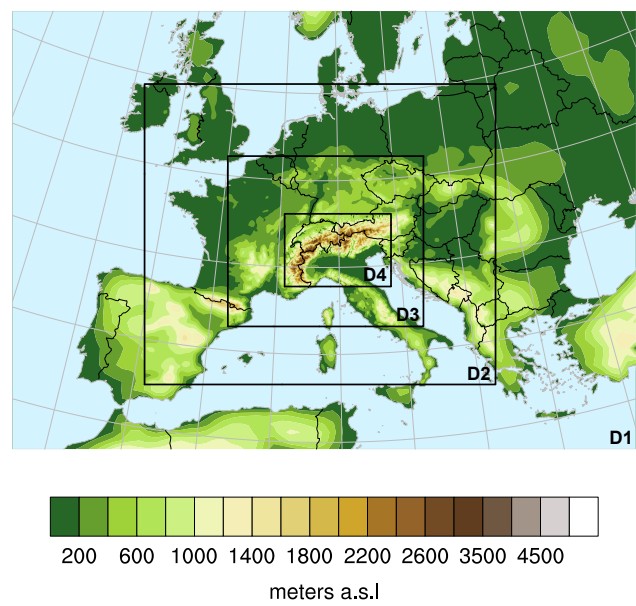

**Figure 1.** The four nested domains in used in the dynamical downscaling.

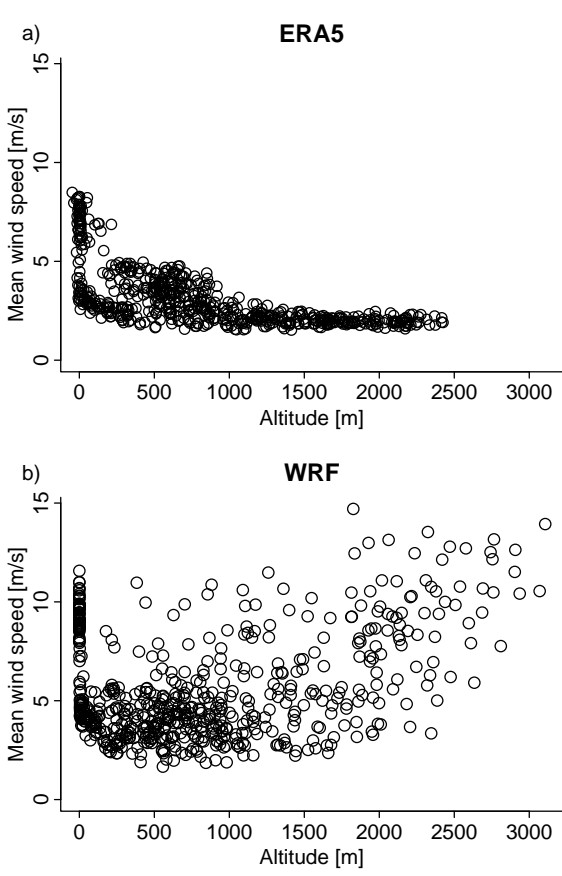

**Figure 2.** Relationship between mean winter wind speed against altitude for ERA5 (a) and the the WRF model (ERAI-WRF simulation) (b).

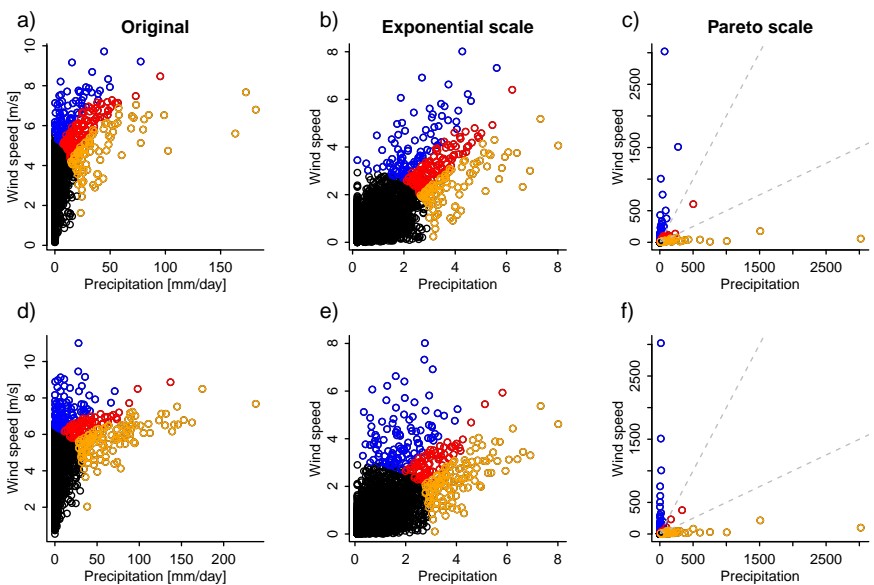

**Figure 3.** Scatterplots of daily precipitation and wind speed in November-March (1980-1999) for the location with the highest tail dependence $\chi$ ($q = 0.95$) in the ERAI-WRF simulations (a-c). CESM-WRF simulations for the same location are shown in (d-f). Shown are the original values (a and d), after transformation into exponential marginals (b and e) and after transformation into Pareto marginals (c and f). The colors highlight the three separating sets $W$ to compute the KL divergence, see Eq. (1), for a high threshold (see main text). In c) and f), the three sets are separated by dashed lines.

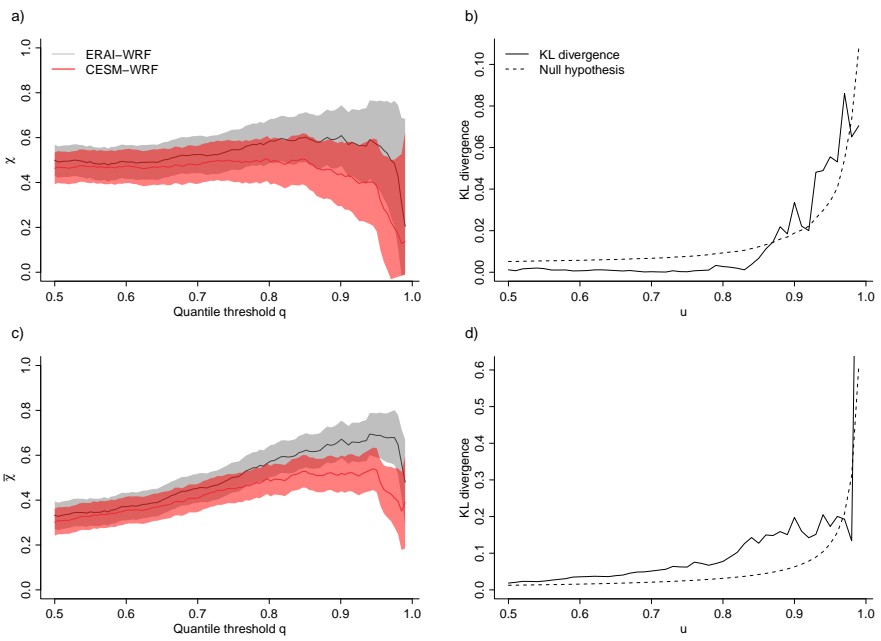

**Figure 4.** Illustration of the distance metrics between bivariate tails for the location with highest estimated tail dependence $\chi$ at $q = 0.95$ in ERAI-WRF. Left: Tail dependence parameters $\chi$ (a) and $\bar{\chi}$ (c) for daily precipitation sums and daily maximum wind speed for different quantile-based thresholds $q$. Shading highlights the 95% confidence intervals. Grey: ERAI-WRF. Red: CESM-WRF. Right: Two different Kullback–Leibler (KL) divergences (eq. (1) with $W = 5$) for the tails of the bivariate precipitation-wind speed distribution between ERAI-WRF and CESM-WRF (solid lines). Dashed lines highlight the 95% confidence interval of the null hypothesis assuming an equal dependence structure. b) KL divergence based on the minimum (i.e., $\min(X_1, X_2) > u$). d) KL divergence based on the sum (i.e., $X_1 + X_2 > u$).

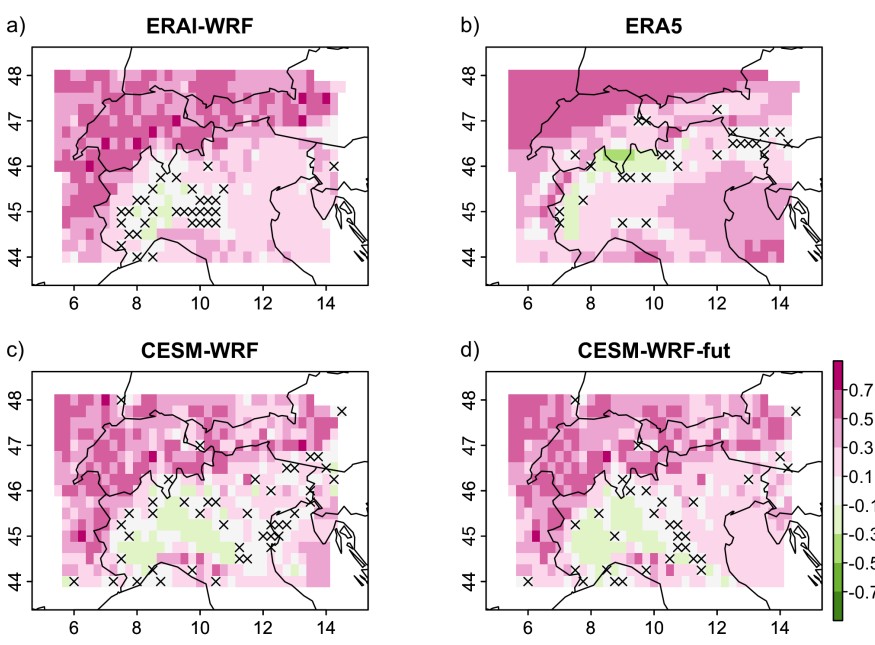

**Figure 5.** Spearman's rank correlation between daily precipitation sums and maximum wind speed in the extended winter (November-March). a) ERAI-WRF, b) ERA5, c) CESM-WRF, d) CESM-WRF-fut. Non-significant correlations ($\alpha = 0.05$) are marked with a cross.

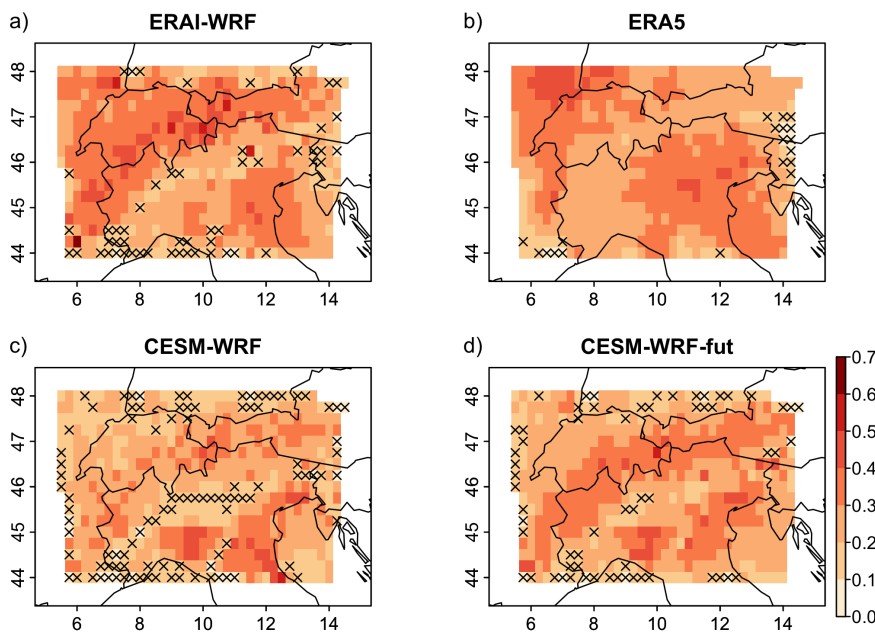

**Figure 6.** Tail dependence ($\chi$ with $q = 0.95$) between daily precipitation sums and maximum wind speed in the extended winter (November-March). Tail dependence was computed considering block maxima over a maximum range 5 days temporally and 1.75 degrees spatially. a) ERAI-WRF, b) ERA5, c) CESM-WRF, d) CESM-WRF-fut. Non-significant values based on bootstraps with the same maximum block size ($\alpha = 0.05$) are marked with a cross.

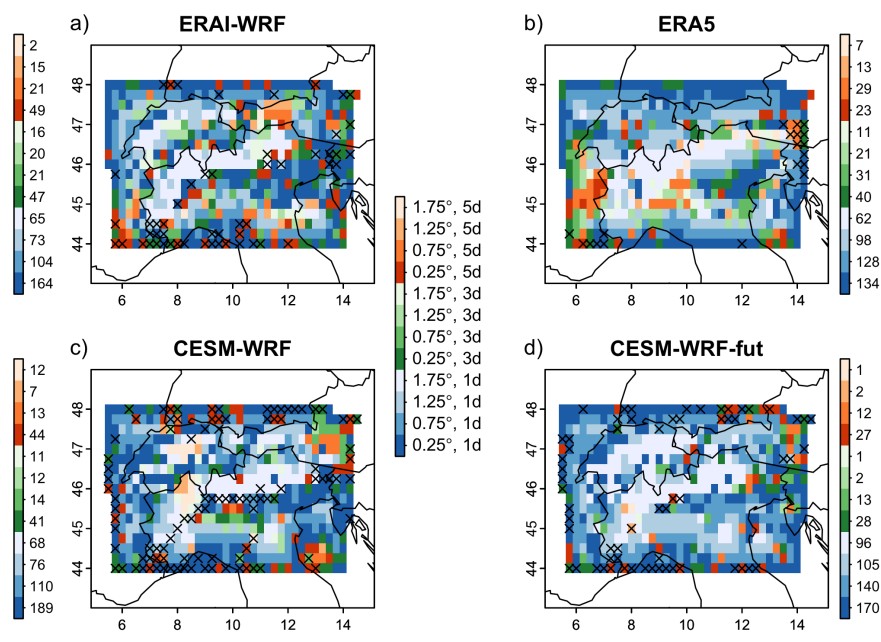

**Figure 7.** Blocks for which the maximum tail dependence ($\chi$ with $q = 0.95$) between daily precipitation sums and maximum wind speed in the extended winter (November-March) is attained (Figure 6). Block sizes range from 0.25°, 1 day to 1.75°, 5 days. Blue, green and orange refer to time lags of 1, 3 and 5 days respectively. Darker shading illustrates higher spatial proximity. The color bars next to the maps show the number of grid points of that color in the corresponding map. a) ERAI-WRF, b) ERA5, c) CESM-WRF, d) CESM-WRF-fut. Grid points with non-significant tail dependence are marked with a cross (see Figure 6).

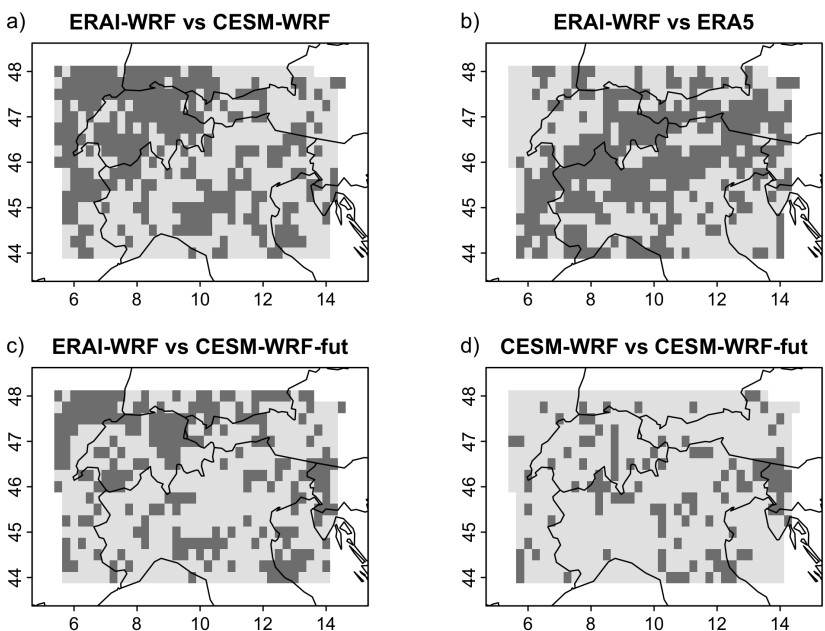

**Figure 8.** Locations for which the dependence between the tails of daily precipitation sums and wind speed maxima is significantly different based on the KL divergence, Eq. (1) with $u = 0.9$ and $W = 5$ (dark grey, with $\alpha = 0.05$). Dependence is assessed for the blocks that attain maximum tail dependence $\chi$ (at $q = 0.95$) (see Figure 6). Shown are comparisons between a) ERAI-WRF and CESM-WRF, b) ERAI-WRF and ERA5, c) CESM-WRF and CESM-WRF-fut, and d) ERAI-WRF and CESM-WRF-fut.

**Table 1.** Sensitivity analysis of KL divergence (eq. (1)). Reported is the fraction of grid points with significantly different ($\alpha = 0.05$) precipitation-wind speed dependence structure between two datasets for different thresholds $u$ (with $W = 5$). The case $u = 0.90$ is shown in Figure 8.

|  | $u = 0.80$ | $u = 0.85$ | $u = 0.90$ | $u = 0.95$ |
|---|---|---|---|---|
| ERAI-WRF vs CESM-WRF | 0.61 | 0.54 | 0.46 | 0.32 |
| ERAI-WRF vs ERA5 | 0.59 | 0.53 | 0.49 | 0.40 |
| ERAI-WRF vs CESM-WRF-fut | 0.53 | 0.45 | 0.36 | 0.27 |
| CESM-WRF vs CESM-WRF-fut | 0.30 | 0.23 | 0.18 | 0.19 |

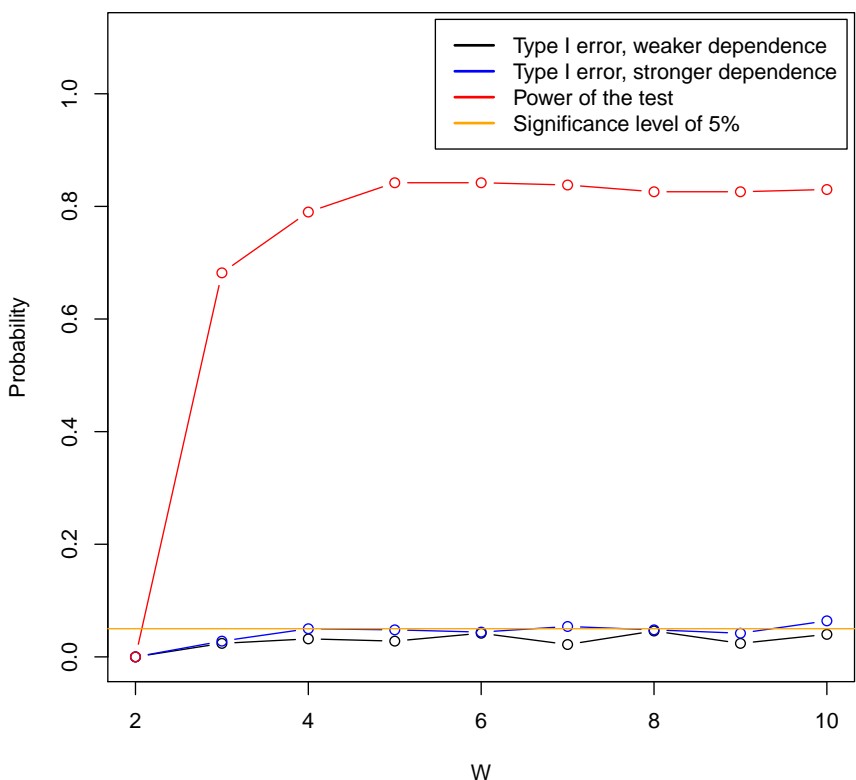

**Figure A1.** Simulation study using empirical margins.

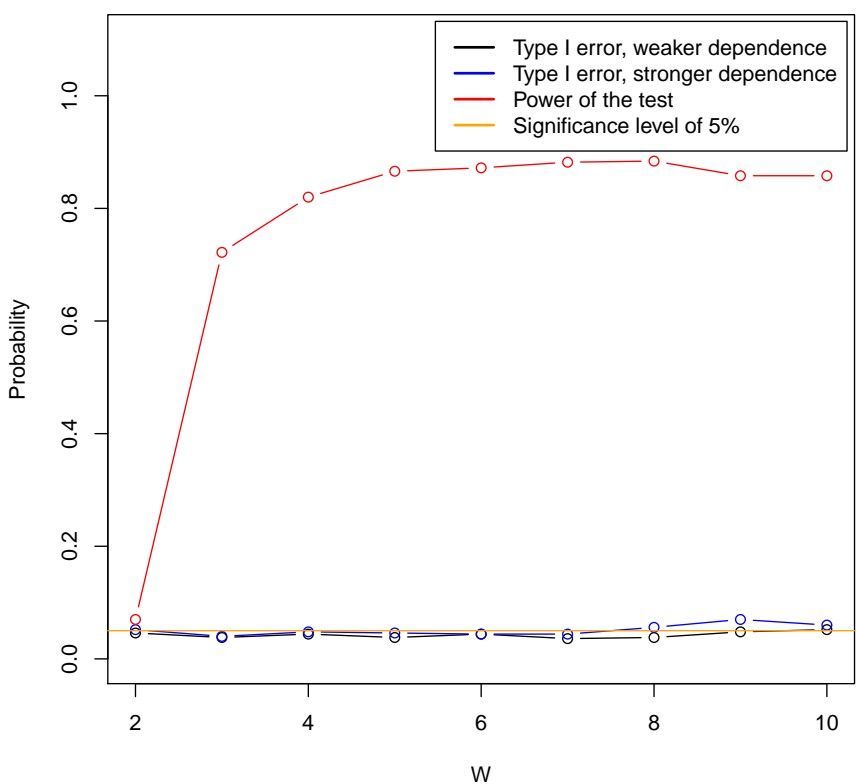

**Figure A2.** Simulation study using true margins.