# Peer review of "Evaluating the dependence structure of compound precipitation and wind speed extremes"

_Earth System Dynamics, 2020_

## Referee Comment (RC1) · 15 Jun 2020

General comments:

The manuscript presents a new methodological tool to compare compound extreme distributions between different datasets. The ability of a model to reproduce the behavior of compound extremes is of fundamental importance to assess climate related risks and to predict the evolution of such compound extreme events with climate change. The new metric is based on the Kullback-Leibler divergence. It is tested on different pairs of models and allows the comparison between different models regarding compound extreme distributions.

[Figure]

Âă I find the manuscript well-motivated and clearly written, even for non-specialists of climate. The new metric seems promising and the statistical analysis made with it is well described and seems solid. The interpretation of the results is convincing to me, although my knowledge of climate models is limited.

ÂăSpecific comments:

It is not mentioned whether the results are stable against different partitioning of the extremal region.ÂăYou could add a few words about it : Are there partitions that are more suited than others? What made you chose this particular partition?

Technical corrections:

-l 29 : 2 times the word 'studies' -l 144 : behavior -l 240 : may result

---

## Referee Comment (RC2) · Anonymous Referee #2 · 2 Jul 2020

This manuscript compares the dependence structure of compound precipitation and wind speed extremes in different sets of data: the ERA5, the dynamically downscaled ERA-Interim using the regional WRF model, the dynamically downscaled CESM with present conditions using the WRF model, and also a dynamically downscaled CESM run for the future. The technique used is an advanced statistical technique on bivariate asymptotic tail dependence. This is an interesting study which deserves publication in ESD. I have a few points on the interpretation of the results, and the limits of this study, that the authors may consider.

First, it seems to me strange to study extremes in a nudged system (ERAI-WRF). This

means that there is a modification of the dynamical equations of WRF and the extremes could then be biased. First could you run it without the nudging, and if not this should appear somewhere in your interpretation or conclusions.

A second aspect is the fact that the domain over which dowscaling is done looks small (no information provided on this by the way on the specific configuration of running WRF). This should have considerable impact on the extremes in particular for wind but also for precipitation. There were a lot of work done in this context at the beginning of the 21th century on that topic, showing that small domains are considerably constraining the internal dynamics of the regional model, and hence all the statistical properties within the model. This should also play an important role and should be discussed in the conclusions or in the interpretation of the results.

In Figure 1. It would be nice to see the observations too.

---

## Short Comment (SC1) · 19 Aug 2020

The manuscript titled "Evaluating the dependence structure of compound precipitation and wind speed extremes" aims to estimate the likelihood of compound precipitation and wind speed extremes. In particular the Authors use metrics (the coefficients $\chi$, $\overline{\chi}$, and KL measure of divergence) to measure the tail of bivariate distributions, and if it is similar between different datasets. The Authors use data from one reanalysis product (ERA5) and three model simulations (ERAI-WRF, CESM-WRF, CESM-WRF-fut) over a period of 20 years.

General comment: The manuscript is well written. The methodology is well grounded,

and of interest also for other compound events. The conclusions concerning the ERA5, which is considered as state-of-the-art dataset, are of value. Thus, I think that the manuscript should be accepted after minor revision.

Here you have some specific questions/comments:

- Application of the methodology to real data: Have the Authors an idea about the minimum number of data necessary to obtain a reliable estimate of the proposed metrics? Some details about this could be very useful.

- Please give details about how you have calculated the KL measure of divergence, similar to the information given for the calculation of the coefficients $\chi, \overline{\chi}$.

- A reference for the risk function could be useful.

- I think that in Figure 7 "K=3" should be substituted with "W=3", to be coherent with the text. Similar comment applies to the caption of Table 1.

- Line 261 change "bivarate" with "bivariate".

—————————————————————

---

## Referee Comment (RC3) · Anonymous Referee #3 · 25 Aug 2020

**Disclaimer:**
Even though I read the whole paper and appreciated both the methodological and applied aspects of this research, my review mostly revolves around the statistical contributions of this paper, which I'm more confident to comment on.

**Summary:**
In this paper, the authors propose a new statistical metric to compare the bivariate joint tails of two different datasets. This metric, which relies on the Kullback-Leibler (KL) divergence based on the count of points in certain number of "extreme sets", provides a single number that can be used to assess whether or not the joint tails are different,

and if so, by how much they differ. It is proposed as being complementary to more classical measures, such as the $\chi$-measure introduced in the paper that is widely used in extreme-value theory. The proposed KL metric depends on the number of sets, $W$, which has to be defined by the analyst, and is "non-parametric" in the sense that it does not rely on stringent model assumptions. In the paper, the proposed metric is used to estimate the likelihood of compound precipitation and wind speed extremes derived from different climate model outputs.

**General assessment and general comments:**
In my opinion, the paper is well written and concise with interesting practical results. Methodologically, the proposed metric is well-grounded but is not particularly novel as the Kullback-Leibler divergence (here based on the multinomial distribution) has been used extensively in other areas of statistics. The novelty probably relies on its specific use to study bivariate extremes and to compare bivariate joint tails of extreme precipitation and wind speed, although it is based on a previously published paper by one of the authors (Naveau et al., 2014, JRSS B). Overall, I like the paper and find the results quite interesting, yet several questions remain unanswered. My general and specific comments below mostly focus on the statistical contributions of the paper.

1. The $\chi$ measure is computed based on "local block maxima". I think it is easier to understand what the $\chi$ and $\bar{\chi}$ measures represent when used with the original daily data, rather than with block maxima. With original data, if $\chi = 0$ this implies asymptotic independence of daily data, but how should we interpret it with block maxima? It would be good to add a few lines or a short paragraph to better explain the statistical meaning and the practical interpretation of the proposed metrics ($\chi$, and KL-based) when they are used with block maxima. And why did you choose to compute $\chi$ based on block maxima and not block means or block minima? What is the rational behind this choice? Is it somehow more informative to compare joint tails?

2. A major question that remains unclear to me is what do we gain with the proposed KL measure? As pointed out by the authors on page 5, we could compute a measure

$\chi^{(1)}$ based on the first dataset, and another measure $\chi^{(2)}$ based on the second dataset and compare their values. The authors argue that they want just a single number to assess whether the tails are different and by how much. I get that. But why not simply doing a formal statistical test of whether $\widehat{\chi}^{(1)}$ is statistically different from $\widehat{\chi}^{(2)}$?? The test statistic (or the corresponding p-value) would indeed be a single value that could be used to assess whether the tails are different, and by how much. Moreover, the proposed KL metric is $\chi^2$-squared distributed ASYMPTOTICALLY, while testing for $\chi^{(1)} = \chi^{(2)}$ could—I think—be done EXACTLY for finite $n$ (or be based on the corresponding asymptotic normal distribution). A partial answer to my question above ("what do we gain with the KL measure?") may be that the KL measure is probably more informative for testing whether the joint tails are different because it relies on full distribution of counts within extreme sets, rather than only on information about "the diagonal $F_1(X_1) = F_2(X_2)$"... but without a proper simulation study, this is difficult to claim (especially that the KL measure depends on the choice of $W$). It would be good if the authors could elaborate on that, and complement the paper with a short simulation study to assess the gains of the KL measure compared to a simple test $\chi^{(1)} = \chi^{(2)}$.

3. This point is related to the point 2 above, but I split it into two parts so the authors can more easily address the several questions that I have. Another major question related to the proposed KL measure is how to set the number of extreme sets, $W$, to use. In the paper the authors choose $W = 3$, but there is no optimality with this choice. In fact, while the proposed KL measure is not well-defined when at least one of the sets is empty, the more classical $\chi$-measure is always well defined (so testing $\chi^{(1)} = \chi^{(2)}$ is always possible). This is a major drawback of the KL measure, I think, since under asymptotic independence we should EXPECT that the probability mass will concentrate on the axes (on the Pareto scale) with no point in the interior (so extreme sets should be empty in the limit!). Of course, in practice, there will always be points in the interior and ways to ensure that the extreme sets are non-empty, but it still raises the question of how to choose the number of sets $W$ and the sets themselves. A related question is what is the efficiency of the statistical procedure for different numbers of

sets, $W$? In my opinion, it would be good to complement the paper with a simulation study, in order to investigate this issue in more details and come up with some concrete advice for practitioners on the selection of sets. Is there an "automatic" way to do this "well"?

4. Another major point that is unclear to me is the treatment of marginal distributions. I assume that margins are estimated non-parametrically (with ranks) to compute the $\chi$-measure, and that the extreme sets are defined based on data transformed to a common scale (e.g., Pareto), but there is no mention of marginal modeling in the paper. Does it matter here, since the KL-measure is non-parametric? I think this should be clarified. Marginal modeling usually has a major effect on the final results and their interpretation, so care is needed. In particular, how was the uncertainty related to marginal modeling taken into account (if it was)? The authors mention a bootstrap procedure for the $\chi$-measure, but does it take marginal estimation uncertainty into account or does it only account for the estimation of the dependence structure?

5. Figures 5-6: Even if I understand why the authors chose different block sizes (i.e., spatial lags and temporal windows), I find it difficult to interpret the results in Figure 5 given that the color in each pixel represents the tail dependence of potentially completely different events based on different block sizes. This may also explain why the figure looks a bit "noisy". Wouldn't it make more sense to produce such a map for each block size separately, and then present only the "most relevant" one (or potentially 2 block sizes of interest)? In my opinion, this would be much easier to interpret.

6. Although the authors cite relevant papers related to extreme-value theory, some general review papers (or classical textbooks) could be added in my opinion to help non-experts navigate through this extensive literature.

**Specific comments:**
1. Page 2, Line 29: "studies studies"
2. Page 5, Line 119: If I'm not mistaken, the $\bar{\chi}$ measure has been introduced in a paper

by Coles, Heffernan and Tawn (1999) published in Extremes, not by Ledford and Tawn (1996). Please add this reference.

3. Page 5: Line 126 says "inspect their behavior as $q \to 1$" but Line 128 says "We generally estimate $\chi$ at $q = 0.95$". I agree and I get what the authors want to say, but these two sentences sound a bit contradictory. Please reformulate.

4. Page 5, Lines 149-150: you mention the sum and the minimum as the risk function $r(x)$. Why not considering the maximum, as well, which is perhaps more commonly used than the minimum?

5. Page 6, Line 155: write "$A_w^{(j)}$" instead of "$A_W$"?

6. Page 6, Line 164, "The statistic $d_{12}$ follows a $\chi^2(W-1)$ distribution is the limit": Do you mean "in the limit as $n \to \infty$"? Also, is this valid under the null hypothesis that the tails are the same? Please clarify.

7. Page 6, Lines 181-182, "$q = 0.95$ and $u = 0.9$": why did you choose different numbers? Does it matter?

8. Page 7, Line 199: write "In particular, in the south of the Alps" (add "in the")

9. Page 7, Line 213-215: Table 1 shows the results are different as $u$ increases. What do you conclude? And what if $q$ increases?

10. Page 8, Line 224: write "Because the model setting determines the dependence structure" (add "the")

11. Page 8, Lines 228-229: the sentence "This is to ensure ... (e.g., Foehn)" sounds odd to me. Please consider rewriting.

12. Figure 3: The difference in tail behavior for the two datasets from $q = 0.8$ is already quite clear based on the $\chi$-measure. This comes back to my general comments above: do we really need the new KL metric to detect this?
* * *

---

## Author Comment (AC2) · 15 Sep 2020

This manuscript compares the dependence structure of compound precipitation and wind speed extremes in different sets of data: the ERA5, the dynamically downscaled ERA-Interim using the regional WRF model, the dynamically downscaled CESM with present conditions using the WRF model, and also a dynamically downscaled CESM run for the future. The technique used is an advanced statistical technique on bivariate asymptotic tail dependence. This is an interesting study which deserves publication in ESD. I have a few points on the interpretation of the results, and the limits of this study, that the authors may consider.

[Figure]

*Thank you.*

First, it seems to me strange to study extremes in a nudged system (ERAI-WRF). This means that there is a modification of the dynamical equations of WRF and the extremes could then be biased. First could you run it without the nudging, and if not this should appear somewhere in your interpretation or conclusions.

*The reviewer is correct that the ERAI-WRF is nudged to the driving reanalysis ERA Interim. The reason for this is that the simulation should stay close to large-scale behavior of the reanalysis data. As mentioned in the manuscript, we only use wind, temperature and humidity above the planetary boundary layer and the nudging is not strong. So, we agree that to some extent the behavior of extremes might be changed due to the modification of the dynamical equations, but we think that this effect is minor. We also would like to point out that the precipitation is not nudged. To quantify the effect of nudging (and show that the effect is minor) a second simulation would be helpful, but currently we do not have the computational resources to perform such a simulation. To clarify this, in the revision we will comment on this aspect in our discussion of the results related to the ERAI-WRF simulation.*

A second aspect is the fact that the domain over which downscaling is done looks small (no information provided on this by the way on the specific configuration of running WRF). This should have considerable impact on the extremes in particular for wind but also for precipitation. There were a lot of work done in this context at the beginning of the 21th century on that topic, showing that small domains are considerably constraining the internal dynamics of the regional model, and hence all the statistical properties within the model. This should also play an important role and should be discussed in the conclusions or in the interpretation of the results.

*We see that the manuscript was not clear regarding the setup. We only show the innermost domain of a nested regional climate modelling approach using four nests in total. Domain 1 spans over Europe. The regions of the four nests are illustrated in the*

[Figure]

*figure below and we will include this figure in the revised manuscript. In addition, we will add the following explanation: "The horizontal resolution of the four two-way nested domains (Fig. 1) are 54, 18, 6 and 2 km, respectively. The innermost domain 4 covers the box $[4.75E, 15.25E] \times [43.25N, 48.75N]$ and is used in this study exclusively."*

In Figure 1. It would be nice to see the observations too.

*This is a pure modelling study and we do not have observational data for wind speed at different height at hand. This is why we refer to published results.*

—————————————————

[Figure]

Fig. 1. The four nested domains used in the dynamical downscaling.

200   600   1000  1400  1800  2200  2600  3500  4500

meters a.s.l

---

## Author Comment (AC3) · 15 Sep 2020

*We thank Carlo de Michele for his constructive comments on our manuscript.*

The manuscript titled "Evaluating the dependence structure of compound precipitation and wind speed extremes" aims to estimate the likelihood of compound precipitation and wind speed extremes. In particular the Authors use metrics (the coefficients chi, chibar, and KL measure of divergence) to measure the tail of bivariate distributions, and if it is similar between different datasets. The Authors use data from one reanalysis product (ERA5) and three model simulations (ERAI-WRF, CESM-WRF, CESM-WRF-fut) over a period of 20 years.

General comment: The manuscript is well written. The methodology is well grounded and of interest also for other compound events. The conclusions concerning the ERA5, which is considered as state-of-the-art dataset, are of value. Thus, I think that the manuscript should be accepted after minor revision.

*Thank you.*

Here you have some specific questions/comments:

- Application of the methodology to real data: Have the Authors an idea about the minimum number of data necessary to obtain a reliable estimate of the proposed metrics? Some details about this could be very useful.

*This depends in general on the distribution of the data. For the chi coefficient we provide confidence intervals. For the KL statistics confidence interval can be obtained via simulation studies depending on the underlying distribution. In the revision we will add a comment in this regard.*

- Please give details about how you have calculated the KL measure of divergence, similar to the information given for the calculation of the coefficients chi, chibar.

*For computing the KL divergence, equation 1 and the one above provide the necessary equations to compute it.*

- A reference for the risk function could be useful.

*The sum is a classical way of looking at extremes, which treats all variables equally. If one projects all points that are large in this risk measure on the sphere $r(x) = 1$, then one obtains the so-called spectral measure, a popular object in extreme value theory. The minimum on the other hand excludes the axes and is therefore also suitable for asymptotically independent data. We will add some explanation regarding the choice of risk functions in the revision.*

- I think that in Figure 7 "K=3" should be substituted with "W=3", to be coherent with

the text. Similar comment applies to the caption of Table 1.

*Thanks.*

- Line 261 change "bivarate" with "bivariate".

*Thanks.*

---

## Author Comment (AC4) · 15 Sep 2020

**Disclaimer:**

Even though I read the whole paper and appreciated both the methodological and applied aspects of this research, my review mostly revolves around the statistical contributions of this paper, which I'm more confident to comment on.

*We highly appreciate the constructive comments.*

**Summary:**

In this paper, the authors propose a new statistical metric to compare the bivariate joint

tails of two different datasets. This metric, which relies on the Kullback-Leibler (KL) divergence based on the count of points in certain number of "extreme sets", provides a single number that can be used to assess whether or not the joint tails are different, and if so, by how much they differ. It is proposed as being complementary to more classical measures, such as the $\chi$-measure introduced in the paper that is widely used in extreme-value theory. The proposed KL metric depends on the number of sets, W, which has to be defined by the analyst, and is "non-parametric" in the sense that it does not rely on stringent model assumptions. In the paper, the proposed metric is used to estimate the likelihood of compound precipitation and wind speed extremes derived from different climate model outputs.

**General assessment and general comments:**

In my opinion, the paper is well written and concise with interesting practical results. Methodologically, the proposed metric is well-grounded but is not particularly novel as the Kullback-Leibler divergence (here based on the multinomial distribution) has been used extensively in other areas of statistics. The novelty probably relies on its specific use to study bivariate extremes and to compare bivariate joint tails of extreme precipitation and wind speed, although it is based on a previously published paper by one of the authors (Naveau et al., 2014, JRSS B). Overall, I like the paper and find the results quite interesting, yet several questions remain unanswered. My general and specific comments below mostly focus on the statistical contributions of the paper.

*Thank you. Note that Naveau et al. 2014 only treated univariate times series, not bivariate (compound) events.*

1. The $\chi$ measure is computed based on "local block maxima". I think it is easier to understand what the $\chi$ and $\bar{\chi}$ measures represent when used with the original daily data, rather than with block maxima. With original data, if $\chi = 0$ this implies asymptotic independence of daily data, but how should we interpret it with block maxima? It would be good to add a few lines or a short paragraph to better explain the statistical mean-

ing and the practical interpretation of the proposed metrics ($\chi$, and KL-based) when they are used with block maxima. And why did you choose to compute $\chi$ based on block maxima and not block means or block minima? What is the rationale behind this choice? Is it somehow more informative to compare joint tails?

*Taking block maxima is motivated by the underlying scientific question. We are interested in (correlated) extremes in precipitation and wind, which might not occur at the same time or the same location but are driven by the same atmospheric process. These events can still cause disproportionate impacts. Furthermore, thresholding maxima implies that $\chi$, $\bar{\chi}$ and KL really measure very extremal upper tail behaviour. The drawback is the smaller sample size. However, this effect will also happen with thresholding, see e.g. Ferreira, A. and de Haan, L. (2015). On the block maxima method in extreme value theory: PWM estimators. The Annals of Statistics, 43(1):276–298.*

*Block maxima (instead of means or minima) are chosen because the interest is in the dependence between positive extremes of precipitation and wind speed.*

2. A major question that remains unclear to me is what do we gain with the proposed KL measure? As pointed out by the authors on page 5, we could compute a measure $\chi^{(1)}$ based on the first dataset, and another measure $\chi^{(2)}$ based on the second dataset and compare their values. The authors argue that they want just a single number to assess whether the tails are different and by how much. I get that. But why not simply doing a formal statistical test of whether $\chi^{(1)}$ is statistically different from $\chi^{(2)}$?? The test statistic (or the corresponding $p$-value) would indeed be a single value that could be used to assess whether the tails are different, and by how much. Moreover, the proposed KL metric is $\chi^2$-squared distributed ASYMPTOTICALLY, while testing for $\chi^{(1)} = \chi^{(2)}$ could– I think–be done EXACTLY for finite $n$ (or be based on the corresponding asymptotic normal distribution). A partial answer to my question above ("what do we gain with the KL measure?") may be that the KL measure is probably more informative for testing whether the joint tails are different because it relies on full distribution of counts within extreme sets, rather than only on information about "the diagonal $F_1(X_1) = F_2(X_2)$"...

but without a proper simulation study, this is difficult to claim (especially that the KL measure depends on the choice of $W$). It would be good if the authors could elaborate on that, and complement the paper with a short simulation study to assess the gains of the KL measure compared to a simple test $\chi^{(1)} = \chi^{(2)}$.

*We appreciate the comment and suggestions. However, such a simulation study would go much beyond the interest of the readership of ESD. What one could say without a simulation study is that if we consider two models with the same $\chi$ coefficient but different dependence structure, then it is impossible to distinguish the two cases with the easier test the reviewer proposes. Thus the referee is right when they stated that $\chi$ focuses on the "diagonal". Furthermore, in this work we focus on the bivariate case, but the KL estimate defined by equation (1) could be easily implemented with higher dimensions $d = 3, 4, \ldots$, because it is just based on counting points in different subsets. With $\chi$ coefficients, the number of pairs will increase rapidly with $d$. In addition, $\chi$ coefficients will only capture pairwise dependencies. The KL does not have this problem and can easily be used for trivariate compounds events. We will add these explanations in the revision to better motivate the usage of the KL divergence as a difference measure.*

3. This point is related to the point 2 above, but I split it into two parts so the authors can more easily address the several questions that I have. Another major question related to the proposed KL measure is how to set the number of extreme sets, W, to use. In the paper the authors choose $W = 3$, but there is no optimality with this choice. In fact, while the proposed KL measure is not well-defined when at least one of the sets is empty, the more classical $\chi$-measure is always well defined (so testing $\chi^{(1)} = \chi^{(2)}$ is always possible). This is a major drawback of the KL measure, I think, since under asymptotic independence we should EXPECT that the probability mass will concentrate on the axes (on the Pareto scale) with no point in the interior (so extreme sets should be empty in the limit!). Of course, in practice, there will always be points in the interior and ways to ensure that the extreme sets are non-empty, but it still raises the

question of how to choose the number of sets W and the sets themselves. A related question is what is the efficiency of the statistical procedure for different numbers of sets, $W$ ? In my opinion, it would be good to complement the paper with a simulation study, in order to investigate this issue in more details and come up with some concrete advice for practitioners on the selection of sets. Is there an "automatic" way to do this "well"?

*At some level, $\chi$ is also based on an arbitrary choice because it is based on counting the number of points in the very specific "upper corner" ($X_1 > u, X_2 > u$), given $X_1 > u$. Our proposed KL divergence introduces more flexibility in terms of the choosing the norm, the number of set and the shape of sets. If the conditioning norm was equal to $r(x) = \min(x_1, 0)$ and the partition just one set, $W = \{X_1 > u, X_2 > u\}$, then the KL measure will contain the same information than $\chi$. Hence, instead of being a competitor, the KL measure broadens the scope of $\chi$ coefficients and allows for more detailed analysis. Of course, this added flexibility leads to more choices.*

*The case of asymptotic independence can be covered by our KL using $r(x) = \min(x_1, x_2)$ and by choosing sets $W_i$ such that the probabilities of being no-empty in each set is positive. By assuming a second order type condition (classical multivariate EVT), Engelke, Naveau and Zhou (in prep) show that the convergence of our KL estimate towards a Chi-square distribution is still valid. For this theoretical statement, the marginals were supposed to be unknown with possibly different shape parameters. Hence, rank-based transforms were used, this answers the point 4 raised by the referee. Under asymptotic dependence the empirical marginal normalization does have an effect on the asymptotic distribution. However, this effect is rather minor with little influence on the power of the test and the Type I error, as illustrated by the two figures and the corresponding simulation study below.*

*We simulated $n = 2000$ samples $X^{(1)}$ and $X^{(2)}$ of the outer power Clayton copula, which is in the domain of attraction of the logistic extreme value distribution. We chose the parameters such that the limiting $\chi$ coefficients are 0.4 and 0.55, that is, one model*

*with weaker and one with stronger dependence, respectively. Using the KL divergence for a probability threshold of $u = 0.9$, we compare samples $X^{(1)}/X^{(2)}$ for the settings weak/weak, strong/strong and weak/strong and plot in each case the probability of rejecting the null hypothesis of equal tail dependence structures. Note that the former two cases are in line of the null hypothesis, whereas the latter case does not satisfy the null hypothesis. We do the experiment both for known margins and for empirically normalized margins, and for different numbers of sets $W$ in the KL divergence statistic.*

*The two figures below (Figure 1 and 2) show the Type I error of rejecting the null hypothesis in the case the where we have the same tail dependence based on 500 repetitions of the simulation. For both normalizations the significance level of $5\%$ is in general well attained throughout all numbers of sets. The figures also contain the power of the test when the tail dependence structures are different. After $W = 5$ the power stabilises and it seems to decrease slightly when the number of sets is chosen to large. Note that this is only one particular simulation setup and the results on the optimal number of sets can change depending on sample size and strength of tail dependence. We will add these simulation results as appendix to the revised manuscript. Based on these simulation results, we use $W = 5$ in the revised version of the manuscript, which leads to a slightly higher number of significant KL divergences in Figure 7 and Table 1 but otherwise does not affect our main conclusions.*

4. Another major point that is unclear to me is the treatment of marginal distributions. I assume that margins are estimated non-parametrically (with ranks) to compute the $\chi$-measure, and that the extreme sets are defined based on data transformed to a common scale (e.g., Pareto), but there is no mention of marginal modeling in the paper. Does it matter here, since the KL-measure is non-parametric? I think this should be clarified. Marginal modeling usually has a major effect on the final results and their interpretation, so care is needed. In particular, how was the uncertainty related to marginal modeling taken into account (if it was)? The authors mention a bootstrap procedure for the $\chi$-measure, but does it take marginal estimation uncertainty into account

or does it only account for the estimation of the dependence structure?

*The marginals have been transformed to Pareto scale through ranking. As stated in the response to 3., convergence of the KL estimates does not depend on this choice in the case of asymptotic independence. Under asymptotic dependence, empirical marginal normalization does change the asymptotic distribution but with only a very small effect on the robustness of the test, see the simulation study in response to comment 3. We will add the information how margins were transformed into Pareto scale in the revision.*

5. Figures 5-6: Even if I understand why the authors chose different block sizes (i.e., spatial lags and temporal windows), I find it difficult to interpret the results in Figure 5 given that the color in each pixel represents the tail dependence of potentially completely different events based on different block sizes. This may also explain why the figure looks a bit "noisy". Wouldn't it make more sense to produce such a map for each block size separately, and then present only the "most relevant" one (or potentially 2 block sizes of interest)? In my opinion, this would be much easier to interpret.

*We agree that spatial points cannot be directly compared here as they might be based on different block sizes (as indicated in Figure 6). We believe however, that the "noisiness" is an actual signal, related to the extremely high resolution of the original data-generating process (2km) and the complex topography in the alps. This is supported by subpanel b), which is the only one based on much more coarse resolution data (approx. 25km), and consequently shows much smoother spatial gradients (both in Figure 5 and 6). The choice of the block sizes is well motivated by the underlying scientific question (see response to main comment 1 above).*

6. Although the authors cite relevant papers related to extreme-value theory, some general review papers (or classical textbooks) could be added in my opinion to help non-experts navigate through this extensive literature.

*Thank you. We will add the following key references on univariate and bivariate ex-*

*tremes to the manuscript:*

*Embrechts et al., 1997, Modelling Extremal Events: for Insurance and Finance (Springer)*

*Katz et al., 2002, Statistics of extremes in hydrology (AWR 25, 1287-1304)*

*Davison and Huser, 2015, Statistics of Extremes (Annu. Rev. Statistics Appl. 2, 203-235)*

*Engelke and Ivanovs, 2021, Sparse Structures for Multivariate Extremes (Annu. Rev. Statistics Appl., in press)*

**Specific comments:**

1. Page 2, Line 29: "studies studies"

*Thanks.*

2. Page 5, Line 119: If I'm not mistaken, the $\bar{\chi}$ measure has been introduced in a paper by Coles, Heffernan and Tawn (1999) published in Extremes, not by Ledford and Tawn (1996). Please add this reference.

*Thanks, will be done.*

3. Page 5: Line 126 says "inspect their behavior as $q \rightarrow 1$" but Line 128 says "We generally estimate $\chi$ at $q = 0.95$". I agree and I get what the authors want to say, but these two sentences sound a bit contradictory. Please reformulate.

*We will reformulate the second sentence as "To estimate $\chi$ empirically we use a high quantile for which still a reasonable large number of data are available. For these reasons we generally estimate $\chi$ at $q = 0.95$."*

4. Page 5, Lines 149-150: you mention the sum and the minimum as the risk function $r(x)$. Why not considering the maximum, as well, which is perhaps more commonly used than the minimum?

*The sum or the maximum give similar results as they are both used for asymptotically dependent data. The minimum covers also asymptotic independence, and we have included it for this reason.*

5. Page 6, Line 155: write "$A_{w(j)}$" instead of "$A_w$"?

*Yes, thanks.*

6. Page 6, Line 164, "The statistic $d_{12}$ follows a $\chi^2(W\text{-}1)$ distribution is the limit": Do you mean "in the limit as $n \to \infty$"? Also, is this valid under the null hypothesis that the tails are the same? Please clarify.

*Yes, this is true under suitable assumptions, e.g., under asymptotic independence (with additional second order conditions) or if the data is multivariate regularly varying with the same marginal shape parameters (with additional second order conditions). Furthermore, $n \to \infty$ and $u(n) \to 1$ need to converge at the right speed.*

7. Page 6, Lines 181-182, "$q = 0.95$ and $u = 0.9$": why did you choose different numbers? Does it matter?

*These are somewhat arbitrary choices. We have carried out a sensitivity test for different values of $u$, which is shown in Table 1. Qualitatively the pictures doesn't change much (including its scientific interpretation) though of course the numbers are slightly different. In particular, with higher $u$, the number of significant KL divergences decreases, as is expected due to the smaller sample size.*

8. Page 7, Line 199: write "In particular, in the south of the Alps" (add "in the")

*Thanks.*

9. Page 7, Line 213-215: Table 1 shows the results are different as $u$ increases. What do you conclude? And what if $q$ increases?

*The individual numbers change somewhat but the ranking within one column stays the same (except the flip of the first 2 rows at $u = 0.95$, but both have a very similar value).*

*The differences shown in row 1 and 3 are generally larger than the difference in row 4. This is the main scientific finding of the study, as also reported in the abstract: "Overall, boundary conditions in WRF appear to be the key factor in explaining differences in the dependence behaviour between strong wind and heavy rainfall between simulations. In comparison, external forcings (RCP8.5) are of second order." We expect a very similar behavior for different values of $q$. We will add a sentence to make this finding more explicit: "In particular, the differences between ERAI-WRF and CESM-WRF and between ERAI-WRF and CESM-WRF-fut are generally larger than the differences CESM-WRF and CESM-WRF-fut, indicating that the main finding, namely that boundary conditions in WRF appear to be the key factor in explaining differences in the dependence behaviour between wind and rainfall extremes, is robust for different parameter values of the difference measure."*

10. Page 8, Line 224: write "Because the model setting determines the dependence structure" (add "the")

*Thanks.*

11. Page 8, Lines 228-229: the sentence "This is to ensure ... (e.g., Foehn)" sounds odd to me. Please consider rewriting.

*We will rewrite this sentence as "This is to ensure that extremes in wind and precipitation are considered together if they emerge from the same atmospheric processes (e.g. Foehn)."*

12. Figure 3: The difference in tail behavior for the two datasets from q = 0.8 is already quite clear based on the $\chi$-measure. This comes back to my general comments above: do we really need the new KL metric to detect this?

*See our responses to the main comments above. Consider also the example where most of the data is above the diagonal in one case and below the diagonal in the other. Both distributions could have similar $\chi$ but the KL divergence would be large.*

*Interactive comment on Earth Syst. Dynam. Discuss., https://doi.org/10.5194/esd-2020-31, 2020.*

[Figure]

**Fig. 1.** Simulation study using the true margins.

[Figure]

**Fig. 2.** Simulation study using empirical margins.

---

## Author Response (AR1)

*We thank all reviewers for their constructive comments on our manuscript. Below follows a point-by-point response to each comment.*

**Reviewer 1 (Theophile Caby)**

**General comments:**
The manuscript presents a new methodological tool to compare compound extreme distributions between different datasets. The ability of a model to reproduce the behavior of compound extremes is of fundamental importance to assess climate related risks and to predict the evolution of such compound extreme events with climate change. The new metric is based on the Kullback-Leibler divergence. It is tested on different pairs of models and allows the comparison between different models regarding compound extreme distributions.

I find the manuscript well-motivated and clearly written, even for non-specialists of climate. The new metric seems promising and the statistical analysis made with it is well described and seems solid. The interpretation of the results is convincing to me, although my knowledge of climate models is limited.

*Thank you.*

**Specific comments:**
It is not mentioned whether the results are stable against different partitioning of the extremal region. You could add a few words about it: Are there partitions that are more suited than others? What made you chose this particular partition?

*That is a very difficult theoretical question and the answer would depend somewhat on the extremal distribution of the two populations. As a rule of thumb, one would like to use as many sets as possible while guaranteeing that they still contain sufficiently many data points for a stable estimation of the probabilities that go into the Kullback-Leibler divergence. In response to RC3, we have conducted a small simulation study that revealed that W ≥ 5 results in a robust test, and have therefore used W = 5 in the revision. This simulation study is now shown in the Appendix. The key results of the paper remain unchanged.*

Technical corrections:
-l 29: 2 times the word 'studies'
-l 144: behavior
-l 240: may result

*Thanks, we have incorporated the changes.*

**Reviewer 2**

This manuscript compares the dependence structure of compound precipitation and wind speed extremes in different sets of data: the ERA5, the dynamically downscaled ERA-Interim using the regional WRF model, the dynamically downscaled CESM with present conditions using the WRF model, and also a dynamically downscaled CESM run for the future. The

technique used is an advanced statistical technique on bivariate asymptotic tail dependence. This is an interesting study which deserves publication in ESD. I have a few points on the interpretation of the results, and the limits of this study, that the authors may consider.

*Thank you.*

First, it seems to me strange to study extremes in a nudged system (ERAI-WRF). This means that there is a modification of the dynamical equations of WRF and the extremes could then be biased. First could you run it without the nudging, and if not this should appear somewhere in your interpretation or conclusions.

*The reviewer is correct that the ERAI-WRF is nudged to the driving reanalysis ERA Interim. The reason for this is that the simulation should stay close to large-scale behavior of the reanalysis data. As mentioned in the manuscript, we only use wind, temperature and humidity above the planetary boundary layer and the nudging is not strong. So, we agree that to some extent the behavior of extremes might be changed due to the modification of the dynamical equations, but we think that this effect is minor. We also would like to point out that the precipitation is not nudged. To quantify the effect of nudging (and show that the effect is minor) a second simulation would be helpful, but currently we do not have the computational resources to perform such a simulation. We have added these explanations as penultimate paragraph in the discussion.*

A second aspect is the fact that the domain over which downscaling is done looks small (no information provided on this by the way on the specific configuration of running WRF). This should have considerable impact on the extremes in particular for wind but also for precipitation. There were a lot of work done in this context at the beginning of the 21th century on that topic, showing that small domains are considerably constraining the internal dynamics of the regional model, and hence all the statistical properties within the model. This should also play an important role and should be discussed in the conclusions or in the interpretation of the results.

*We see that the manuscript was not clear regarding the setup. We only show the innermost domain of a nested regional climate modelling approach using four nests in total. Domain 1 spans over Europe. The regions of the four nests are illustrated in the new Figure 1. In addition, we have added the following explanation: "The horizontal resolution of the four two-way nested domains (Fig. 1) are 54, 18, 6 and 2 km, respectively. The innermost domain 4 covers the box [4.75E,15.25E] × [43.25N,48.75N] and is used in this study exclusively."*

[Figure]

meters a.s.l

*Figure 1: The four nested domains in used in the dynamical downscaling.*

In Figure 1. It would be nice to see the observations too.

*This is a pure modelling study and we do not have observational data for wind speed at different height at hand. This is why we refer to published results.*

**Short comment (Carlo de Michele)**

The manuscript titled "Evaluating the dependence structure of compound precipitation and wind speed extremes" aims to estimate the likelihood of compound precipitation and wind speed extremes. In particular the Authors use metrics (the coefficients $\chi$ , $\bar{\chi}$, and KL measure of divergence) to measure the tail of bivariate distributions, and if it is similar between different datasets. The Authors use data from one reanalysis product (ERA5) and three model simulations (ERAI-WRF, CESM-WRF, CESM-WRF-fut) over a period of 20 years.

General comment: The manuscript is well written. The methodology is well grounded and of interest also for other compound events. The conclusions concerning the ERA5, which is considered as state-of-the-art dataset, are of value. Thus, I think that the manuscript should be accepted after minor revision.

*Thank you.*

Here you have some specific questions/comments:

- Application of the methodology to real data: Have the Authors an idea about the minimum number of data necessary to obtain a reliable estimate of the proposed metrics? Some details about this could be very useful.

*This depends in general on the distribution of the data. For the χ coefficient we provide confidence intervals. For the KL statistics confidence interval can be obtained via bootstrap, and these will generally depend on the underlying distribution.*

- Please give details about how you have calculated the KL measure of divergence, similar to the information given for the calculation of the coefficients χ and χ̄.

*For computing the KL divergence, equation 1 and the one above provide the necessary equations to compute it.*

- A reference for the risk function could be useful.

*The sum is a classical way of looking at extremes, which treats all variables equally. If one projects all points that are large in this risk measure on the sphere r(x) = 1, then one obtains the so-called spectral measure, a popular object in extreme value theory. The minimum on the other hand excludes the axes and is therefore also suitable for asymptotically independent data.*

- I think that in Figure 7 "K=3" should be substituted with "W=3", to be coherent with the text. Similar comment applies to the caption of Table 1.

*Thanks. Note that we use W=5 in the revision.*

- Line 261 change "bivarate" with "bivariate".

*Thanks.*

**Reviewer 3**

**Disclaimer:**
Even though I read the whole paper and appreciated both the methodological and applied aspects of this research, my review mostly revolves around the statistical contributions of this paper, which I'm more confident to comment on.

*We highly appreciate the constructive comments.*

**Summary:**
In this paper, the authors propose a new statistical metric to compare the bivariate joint tails of two different datasets. This metric, which relies on the Kullback-Leibler (KL) divergence based on the count of points in certain number of "extreme sets", provides a single number that can be used to assess whether or not the joint tails are different, and if so, by how much they differ. It is proposed as being complementary to more classical

measures, such as the χ-measure introduced in the paper that is widely used in extreme-value theory. The proposed KL metric depends on the number of sets, W, which has to be defined by the analyst, and is "non-parametric" in the sense that it does not rely on stringent model assumptions. In the paper, the proposed metric is used to estimate the likelihood of compound precipitation and wind speed extremes derived from different climate model outputs.

**General assessment and general comments:**
In my opinion, the paper is well written and concise with interesting practical results. Methodologically, the proposed metric is well-grounded but is not particularly novel as the Kullback-Leibler divergence (here based on the multinomial distribution) has been used extensively in other areas of statistics. The novelty probably relies on its specific use to study bivariate extremes and to compare bivariate joint tails of extreme precipitation and wind speed, although it is based on a previously published paper by one of the authors (Naveau et al., 2014, JRSS B). Overall, I like the paper and find the results quite interesting, yet several questions remain unanswered. My general and specific comments below mostly focus on the statistical contributions of the paper.

*Thank you. Note that Naveau et al. 2014 only treated univariate times series, not bivariate (compound) events.*

1. The χ measure is computed based on "local block maxima". I think it is easier to understand what the χ and χ̄ measures represent when used with the original daily data, rather than with block maxima. With original data, if χ = 0 this implies asymptotic independence of daily data, but how should we interpret it with block maxima? It would be good to add a few lines or a short paragraph to better explain the statistical meaning and the practical interpretation of the proposed metrics (χ, and KL-based) when they are used with block maxima. And why did you choose to compute χ based on block maxima and not block means or block minima? What is the rationale behind this choice? Is it somehow more informative to compare joint tails?

*Taking block maxima is motivated by the underlying scientific question. We are interested in the relationship between (positive) extremes in precipitation and wind, which might not occur at the same time or at the same location but are driven by the same atmospheric process. These events can still cause disproportionate impacts. Furthermore, thresholding maxima implies that chi, chi-bar and KL really measure very extremal upper tail behavior. The drawback is the smaller sample size. However, this effect will also happen with thresholding, see e.g. Ferreira, A. and de Haan, L. (2015). On the block maxima method in extreme value theory: PWM estimators. The Annals of Statistics, 43(1):276–298.*

*Block maxima (instead of means or minima) are chosen because the interest is in the dependence between positive extremes of precipitation and wind speed.*

*We have added this motivation when introducing the block maxima approach in section 3.1 and section 3.2. In addition, we added the following note: "Note however, that this approach leads to different block sizes depending on the location, which makes a direct comparison in space difficult."*

2. A major question that remains unclear to me is what do we gain with the proposed KL measure? As pointed out by the authors on page 5, we could compute a measure $\chi^{(1)}$ based on the first dataset, and another measure $\chi^{(2)}$ based on the second dataset and compare their values. The authors argue that they want just a single number to assess whether the tails are different and by how much. I get that. But why not simply doing a formal statistical test of whether $\chi^{(1)}$ is statistically different from $\chi^{(2)}$?? The test statistic (or the corresponding p-value) would indeed be a single value that could be used to assess whether the tails are different, and by how much. Moreover, the proposed KL metric is $\chi2$-squared distributed ASYMPTOTICALLY, while testing for $\chi^{(1)} = \chi^{(2)}$ could—I think—be done EXACTLY for finite n (or be based on the corresponding asymptotic normal distribution). A partial answer to my question above ("what do we gain with the KL measure?") may be that the KL measure is probably more informative for testing whether the joint tails are different because it relies on full distribution of counts within extreme sets, rather than only on information about "the diagonal F1(X1) = F2(X2)"... but without a proper simulation study, this is difficult to claim (especially that the KL measure depends on the choice of W). It would be good if the authors could elaborate on that, and complement the paper with a short simulation study to assess the gains of the KL measure compared to a simple test $\chi^{(1)} = \chi^{(2)}$.

*We appreciate the comment and suggestion. However, we believe that such a comprehensive simulation study would go beyond the interest of the readership of ESD. What one can say without a simulation study is that if we consider two distributions with the same χ coefficient but different dependence structure, then it is impossible to distinguish the two cases with the easier test the reviewer proposes. The referee is right when they state that χ focuses on the "diagonal". Furthermore, in this work we focus on the bivariate case, but the KL estimate defined by equation (1) could be easily implemented with higher dimensions d=3, 4, …, because it is just based on counting points in different subsets. With chi coefficients, the number of pairs will increase rapidly with d. In addition, chi coefficients will only capture pairwise dependencies. The KL does not have this problem and can easily be used for trivariate compounds events. We have added the following motivation for the KL divergence at the end of section 3.2:*
*"Note that in the bivariate case, a simple approach to quantify the difference in tail dependence would be the difference between $\chi^{(1)}$ and $\chi^{(2)}$. However, for two distributions with the same χ coefficient but different dependence structure, it is impossible to distinguish the two cases. In a way, χ only focuses on the 'diagonal'. Furthermore, while in this work we focus on the bivariate case, the KL divergence defined by (1) could be easily implemented with higher dimensions d=3, 4, …, because it is only based on counting points in different subsets. In contrast, using χ, the number of pairs will increase rapidly with the dimension d. In addition, χ coefficients will only capture pairwise dependencies."*

3. This point is related to the point 2 above, but I split it into two parts so the authors can more easily address the several questions that I have. Another major question related to the proposed KL measure is how to set the number of extreme sets, W, to use. In the paper the authors choose W = 3, but there is no optimality with this choice. In fact, while the proposed KL measure is not well-defined when at least one of the sets is empty, the more classical χ-

measure is always well defined (so testing $\chi(1) = \chi(2)$ is always possible). This is a major drawback of the KL measure, I think, since under asymptotic independence we should EXPECT that the probability mass will concentrate on the axes (on the Pareto scale) with no point in the interior (so extreme sets should be empty in the limit!). Of course, in practice, there will always be points in the interior and ways to ensure that the extreme sets are non-empty, but it still raises the question of how to choose the number of sets W and the sets themselves. A related question is what is the efficiency of the statistical procedure for different numbers of sets, W ? In my opinion, it would be good to complement the paper with a simulation study, in order to investigate this issue in more details and come up with some concrete advice for practitioners on the selection of sets. Is there an "automatic" way to do this "well"?

*At some level, $\chi$ is also based on an arbitrary choice because it is based on counting the number of points in the very specific "upper corner" $(X_1>u, X_2>u)$", given $X_1>u$. Our proposed KL divergence introduces more flexibility in terms of the choosing the norm, the number of set and the shape of sets. If the conditioning norm was equal to $r(x)=min(x_1,0)$ and the partition just one set, $W = \{X_1>u, X_2>u\}$, then the KL measure will contain the same information than chi. Hence, instead of being a competitor, the KL measure broadens the scope of $\chi$ coefficients and allows for more detailed analysis. Of course, this added flexibility leads to more choices.*

*The case of asymptotic independence can be covered by our KL using $r(x) = min(x_1, x_2)$ and by choosing sets $W_i$ such that the probabilities of being no-empty in each set is positive. By assuming a second order type condition (classical multivariate EVT), Engelke, Naveau and Zhou (in prep) show that the convergence of our KL estimate towards a Chi-square distribution is still valid. For this theoretical statement, the marginals were supposed to be unknown with possibly different shape parameters. Hence, rank-based transforms were used, this answers the point 4 raised by the referee.*
*Under asymptotic dependence the empirical marginal normalization does have an effect on the asymptotic distribution. However, this effect is rather minor with little influence on the power of the test and the Type I error, as illustrated by a small simulation study as explained below.*

*We simulated n=2000 samples $X^{(1)}$ and $X^{(2)}$ of the outer power Clayton copula, which is in the domain of attraction of the logistic extreme value distribution. We chose the parameters such that the limiting $\chi$ coefficients are 0.4 and 0.55, that is, one model with weaker and one with stronger dependence, respectively. Using the KL divergence for a probability threshold of u=0.9, we compare the samples of $X^{(1)}$ and $X^{(2)}$ for the dependence settings weak/weak, strong/strong and weak/strong and plot in each case the probability of rejecting the null hypothesis of equal tail dependence structures. Note that the former two cases are in line of the null hypothesis, whereas the latter case does not satisfy the null hypothesis. We do the experiment both for known margins and for empirically normalized margins, and for different numbers of sets W in the KL divergence statistic.*

*The two figures below show the Type I error of rejecting the null hypothesis in the case the where we have the same tail dependence based on 500 repetitions of the simulation. For both normalizations the significance level of 5% is in general well attained throughout all*

*numbers of sets. The figures also contain the power of the test when the tail dependence structures are different. After W=5 the power stabilizes and it seems to decrease slightly when the number of sets is chosen to large. We therefore suggest to take use W=5. Note that this is only one particular simulation setup and the results on the optimal number of sets can change depending on sample size and strength of tail dependence.*

*We have added these simulation results as appendix to the revised manuscript. Based on these simulation results, we now use W=5 in the revised version of the manuscript, which leads to a slightly higher number of significant KL divergences in Figure 8 and Table 1 but otherwise does not affect our main conclusions.*

[Figure]

[Figure]

4. Another major point that is unclear to me is the treatment of marginal distributions. I assume that margins are estimated non-parametrically (with ranks) to compute the χ-measure, and that the extreme sets are defined based on data transformed to a common scale (e.g., Pareto), but there is no mention of marginal modeling in the paper. Does it matter here, since the KL-measure is non-parametric? I think this should be clarified. Marginal modeling usually has a major effect on the final results and their interpretation, so care is needed. In particular, how was the uncertainty related to marginal modeling taken into account (if it was)? The authors mention a bootstrap procedure for the χ-measure, but does it take marginal estimation uncertainty into account or does it only account for the estimation of the dependence structure?

*The marginals have been transformed to Pareto scale through ranking. As stated in the response to 3., convergence of the KL estimates does not depend on this choice in the case of asymptotic independence. Under asymptotic dependence, empirical marginal normalization does change the asymptotic distribution but with only a very small effect on the robustness of the test, see the simulation study in response to comment 3, which has now been added as Appendix A to the manuscript. We have added the information how margins were transformed into Pareto scale at the end of section 3.2.*

5. Figures 5-6: Even if I understand why the authors chose different block sizes (i.e., spatial lags and temporal windows), I find it difficult to interpret the results in Figure 5 given that the color in each pixel represents the tail dependence of potentially completely different events based on different block sizes. This may also explain why the figure looks a bit "noisy". Wouldn't it make more sense to produce such a map for each block size separately, and then present only the "most relevant" one (or potentially 2 block sizes of interest)? In my opinion, this would be much easier to interpret.

*We agree that spatial points cannot be directly compared here as they might be based on different block sizes (as indicated in Figure 6). We believe however, that the "noisiness" is an actual signal, related to the extremely high resolution of the original data-generating process (2km) and the complex topography in the alps. This is supported by subpanel b), which is the only one based on much more coarse resolution data (~25km), and consequently shows much smoother spatial gradients (both in Figure 5 and 6). The choice of the block sizes is well motivated by the underlying scientific question (see response to comment 1 above).*

6. Although the authors cite relevant papers related to extreme-value theory, some general review papers (or classical textbooks) could be added in my opinion to help non-experts navigate through this extensive literature.

*We have added a general paragraph on extreme value theory (univariate and multivariate) at the beginning of section 3.1 referring to the following key literature:*

*Embrechts et al., 1997, Modelling Extremal Events: for Insurance and Finance (Springer)*
*Coles, 2001, An introduction to statistical modeling of extreme values (Springer)*
*Katz et al., 2002, Statistics of extremes in hydrology (AWR 25, 1287-1304)*
*Naveau et al., 2020, Statistical Methods for Extreme Event Attribution in Climate Science, (Annu. Rev. Statistics Appl., 7, 89–110)*
*Davison and Huser, 2015, Statistics of Extremes (Annu. Rev. Statistics Appl. 2, 203-235)*
*Huser and Wadsworth, in press, Advances in Statistical Modeling of Spatial Extremes (Interdisciplinary Reviews Computational Statistics)*
*Engelke and Ivanovs, 2021, Sparse Structures for Multivariate Extremes (Annu. Rev. Statistics Appl., in press)*

**Specific comments:**
1. Page 2, Line 29: "studies studies"

*Thanks.*

2. Page 5, Line 119: If I'm not mistaken, the $\bar{\chi}$ measure has been introduced in a paper by Coles, Heffernan and Tawn (1999) published in Extremes, not by Ledford and Tawn (1996). Please add this reference.

*Thanks, has been changed.*

3. Page 5: Line 126 says "inspect their behavior as $q \to 1$" but Line 128 says "We generally estimate $\chi$ at $q = 0.95$". I agree and I get what the authors want to say, but these two sentences sound a bit contradictory. Please reformulate.

*We reformulated the second sentence as "To estimate $\chi$ empirically we use a high quantile for which a reasonable large number of data are available. For these reasons we generally estimate $\chi$ at $q = 0.95$."*

4. Page 5, Lines 149-150: you mention the sum and the minimum as the risk function r(x). Why not considering the maximum, as well, which is perhaps more commonly used than the minimum?

*The sum or the maximum give similar results as they are both used for asymptotically dependent data. The minimum covers also asymptotic independence, and we have included it for this reason. We have added this explanation when introducing the risk function.*

5. Page 6, Line 155: write "A_w(j)" instead of "A_w "?

*Yes, thanks.*

6. Page 6, Line 164, "The statistic d12 follows a χ2(W − 1) distribution is the limit": Do you mean "in the limit as n → ∞"? Also, is this valid under the null hypothesis that the tails are the same? Please clarify.

*Yes, this is true under suitable assumptions, e.g., under asymptotic independence (with additional second order conditions) or if the data is multivariate regularly varying with the same marginal shape parameters (with additional second order conditions). Furthermore, n → ∞ and u(n) → 1 need to converge at the right speed. We have added "Under suitable assumptions" to make clear that this convergence is conditioned to some general assumptions.*

7. Page 6, Lines 181-182, "q = 0.95 and u = 0.9": why did you choose different numbers? Does it matter?

*These are somewhat arbitrary choices. We have carried out a sensitivity test for different values of u, which is shown in Table 1. Qualitatively the pictures doesn't change much (including its scientific interpretation) though of course the numbers are slightly different. In particular, with higher u, the number of significant KL divergences decreases, as is expected due to the smaller sample size. We have now added here that we conducted a sensitivity analysis with u in {0.8, 0.85, 0.9, 0.95}.*

8. Page 7, Line 199: write "In particular, in the south of the Alps" (add "in the")

*Thanks.*

9. Page 7, Line 213-215: Table 1 shows the results are different as u increases. What do you conclude? And what if q increases?

*The individual numbers change somewhat but the ranking within one column stays the same (except the flip of the first 2 rows at u=0.95, but both have a very similar value). The differences shown in row 1 and 3 are generally larger than the difference in row 4. This is the main scientific finding of the study, as also reported in the abstract: "Overall, boundary conditions in WRF appear to be the key factor in explaining differences in the dependence behaviour between strong wind and heavy rainfall between simulations. In comparison,*

*external forcings (RCP8.5) are of second order." We expect a very similar behavior for different values of q. We have added a sentence to make this finding more explicit: "In particular, the differences between ERAI-WRF and CESM-WRF and between ERAI-WRF and CESM-WRF-fut are generally larger than the differences CESM-WRF and CESM-WRF-fut, indicating that the main finding, namely that boundary conditions in WRF appear to be the key factor in explaining differences in the dependence behaviour between wind and rainfall extremes, is robust for different parameter values of the difference measure."*

10. Page 8, Line 224: write "Because the model setting determines the dependence structure" (add "the")

*Thanks.*

11. Page 8, Lines 228-229: the sentence "This is to ensure ... (e.g., Foehn)" sounds odd to me. Please consider rewriting.

*We have rewritten this sentence as "This is to ensure that extremes in wind and precipitation are considered together if they emerge from the same atmospheric processes (e.g. Foehn)."*

12. Figure 3: The difference in tail behavior for the two datasets from q = 0.8 is already quite clear based on the χ-measure. This comes back to my general comments above: do we really need the new KL metric to detect this?

*See our responses to the main comments above. Consider also the example where most of the data is above the diagonal in one case and below the diagonal in the other. Both distributions could have similar χ but the KL divergence would be large.*

*A marked-up manuscript version follows.*

[revised manuscript text omitted]

---

## Referee Report (RR1)

**Report**

My main question has been anwered and the typos have been corrected. In agreement with my previous report, I consider that this version of the manuscript should be accepted for publication.